# Structural and functional properties of the transporter SLC26A6 reveal mechanism of coupled anion exchange

**David N Tippett[1], Colum Breen[2], Stephen J Butler[2], Marta Sawicka[1], Raimund Dutzler[1]***

[1]Department of Biochemistry, University of Zurich, Zurich, Switzerland; [2]Department of Chemistry, Loughborough University, Loughborough, United Kingdom

**\*For correspondence:**
dutzler@bioc.uzh.ch

**Competing interest:** The authors declare that no competing interests exist.

**Abstract** Members of the SLC26 family constitute a conserved class of anion transport proteins, which encompasses uncoupled transporters with channel-like properties, coupled exchangers and motor proteins. Among the 10 functional paralogs in humans, several participate in the secretion of bicarbonate in exchange with chloride and thus play an important role in maintaining pH homeostasis. Previously, we have elucidated the structure of murine SLC26A9 and defined its function as an uncoupled chloride transporter (Walter et al., 2019). Here we have determined the structure of the closely related human transporter SLC26A6 and characterized it as a coupled exchanger of chloride with bicarbonate and presumably also oxalate. The structure defines an inward-facing conformation of the protein that generally resembles known structures of SLC26A9. The altered anion selectivity between both paralogs is a consequence of a remodeled ion binding site located in the center of a mobile unit of the membrane-inserted domain, which also accounts for differences in the coupling mechanism.

## eLife assessment

This **important** manuscript combines cryo-EM and a suite of **compelling** whole cell and proteoliposome transport assays to establish the mechanism and structure of the full-length human SLC26A6 chloride/bicarbonate exchangers, including the first partial view of the previously unresolved IVS region of an SLC26 STAS domain. In combination with prior studies on additional SLC26 paralogs, including the SLC26A9 paralog initially reported by the same group, the study provides broadly relevant insights into the mechanistic diversity of the SLC26 transporters. This study is of interest to the biophysics community and the field of membrane transport.

## Introduction

Bicarbonate ($HCO_3^-$), the conjugate base of carbonic acid, is a waste product of the citric acid cycle and an important buffer in the intra- and extracellular environment. Its transport across cellular membranes thus constitutes an important mechanism for the control of intracellular pH and the increase of the buffer capacity of bodily fluids and secretions (*Alka and Casey, 2014*). This process is mediated by distinct channels and secondary transporters, the latter comprising paralogs of the SLC4 and SLC26 families. The SLC26 transporters constitute an abundant class of anion transport proteins that are expressed in all kingdoms of life with a considerable degree of sequence conservation (*Alper and Sharma, 2013*; *Dorwart et al., 2008*). Homologs in prokaryotes and plants were classified as symporters, where the uptake of mono- and divalent anions is coupled to the import of either sodium ions or protons (*Geertsma et al., 2015*; *Wang et al., 2019*; *Wang et al., 2021*). However,

the transporters in animals show mechanistic differences. Among the 10 functional SLC26 paralogs in humans, we find an astonishing mechanistic breadth. Family members function as either coupled anion exchangers, anion transporters with channel-like properties and in the case of SLC26A5 (Prestin), as a motor protein in cochlear outer hair cells (*Alper and Sharma, 2013*). Among the paralogs which have retained their transport function, the distinction between coupled and uncoupled transporters is most pronounced. Whereas most family members operate as exchangers of different anions including the monovalent ions chloride (Cl⁻), $HCO_3^-$, iodide (I⁻) and formate and the divalent ions sulfate and oxalate (*Alper and Sharma, 2013*), the protein SLC26A9 functions as fast passive transporter of Cl⁻ (*Dorwart et al., 2007*; *Walter et al., 2019*). Our previous study has characterized the structural and functional properties of the murine ortholog of SLC26A9 (*Walter et al., 2019*). The protein adopts a homodimeric architecture that is representative for the entire SLC26 family. Each subunit consists of a membrane-inserted transport domain (TM) followed by a cytoplasmic STAS domain, which together with the extended N-terminus mediates the bulk of the subunit interactions with few contacts in the membrane-inserted region (*Walter et al., 2019*). The TMs share a protein fold that was initially identified in the prokaryotic transporter UraA (*Lu et al., 2011*), and that was later defined in the prokaryotic homologue SLC26Dg as common architecture for the entire family (*Geertsma et al., 2015*). In this architecture, the TM segregates into two separate motifs that adopt distinct roles during transport, respectively termed 'core' and 'gate' domains. The gate domain forms a contiguous rigid scaffold with the cytosolic STAS domain that remains static during transport. Conversely the mobile core domain harbors a selective anion binding site and acts as mobile unit during transport. This unit alternately exposes the substrate site to different sides of the membrane as expected for a protein working by an elevator-type transport mechanism (*Drew and Boudker, 2016*; *Walter et al., 2019*). Despite its classification as membrane transporter, SLC26A9 mediates large channel-like chloride currents that saturate at high mM concentration and it efficiently discriminates against $HCO_3^-$ (*Dorwart et al., 2007*; *Walter et al., 2019*). Its function as a fast electrogenic chloride transporter and selection against $HCO_3^-$ is astonishing since several other family members have been classified as Cl⁻/ $HCO_3^-$ exchangers (*Alper and Sharma, 2013*). One of the family members sharing this property is the protein SLC26A6 (*Knauf et al., 2001*; *Lohi et al., 2000*). SLC26A6 is widely expressed in different tissues including the heart, kidney, pancreas, and intestine (*Wang et al., 2020*). Besides its function as a $HCO_3^-$ transporter, it also plays an important role in the extrusion of oxalate to prevent the formation of kidney stones (*Clark et al., 2008*; *Jiang et al., 2006*; *Ohana et al., 2013*). Similar to SLC26A9, the protein acts in synergy with the chloride channel CFTR and was proposed to engage in direct interaction and mutual regulation (*Bertrand et al., 2017*; *Dorwart et al., 2008*; *Shcheynikov et al., 2006a*; *Wang et al., 2006*). The stoichiometry of $HCO_3^-$ and oxalate transport by SLC26 is still controversial, as transport of both ions has been described in different studies as either electrogenic or electroneutral (*Chernova et al., 2005*; *Clark et al., 2008*; *Kim et al., 2013*; *Ohana et al., 2009*; *Shcheynikov et al., 2006b*; *Xie et al., 2002*). This question is particularly pertinent in light of the divergent net charge of both substrates, with $HCO_3^-$ being monovalent and oxalate divalent under physiological conditions. Consequently, the structural basis for the mechanistic distinction between the uncoupled SLC26A9 and the coupled SLC26A6 and the basis for their pronounced selectivity, with Cl⁻ acting as common substrate of both proteins, has remained elusive.

To address these open questions, we here present a thorough structural and functional characterization of the human transporter SLC26A6 combining cryo-electron microscopy with electrophysiology and liposomal transport assays. With respect to its overall structure, SLC26A6 closely resembles its paralog SLC26A9, although it shows a distinct transport mechanism. The protein mediates electroneutral Cl⁻/ $HCO_3^-$ and presumable electrogenic Cl⁻/oxalate exchange, which is in stark contrast to the uncoupled Cl⁻ transport of SLC26A9. This is accomplished by an altered ion binding site located in the mobile core domain of the transmembrane transport unit.

## Results

### Functional characterization of SLC26A6

We have previously characterized the transport properties of murine SLC26A9 by patch clamp electrophysiology and found comparatively large and selective Cl⁻ currents. SLC26A9 currents were shown to reverse at the Nernst potential of the conducting anion, which distinguishes the protein as an

uncoupled chloride transporter (*Walter et al., 2019*; *Figure 1A*, *Figure 1—figure supplement 1A*). The high magnitude of these ohmic currents, which lack any time-dependence in response to voltage changes, originate from an unusually active protein that transports anions with channel-like properties (*Figure 1—figure supplement 1A*). SLC26A9 shows a lyotropic permeability sequence that facilitates the transport of $Cl^-$, isothiocyanate, $I^-$ and nitrate but not sulfate or $HCO_3^-$ (*Walter et al., 2019*; *Figure 1A*, *Figure 1—figure supplement 1A*). The inability to conduct $HCO_3^-$ is an exception in a family where most members work as transporters of this abundant anion (*Alper and Sharma, 2013*; *Ohana et al., 2009*). To investigate mechanistic differences between this passive $Cl^-$ transporter and its paralogs functioning as coupled antiporters, we have turned our attention towards the protein SLC26A6, which was proposed to mediate electrogenic $Cl^-/HCO_3^-$ exchange (*Ohana et al., 2009*; *Shcheynikov et al., 2006b*). Hence, we initially attempted to use patch clamp electrophysiology in the whole-cell configuration to characterize its transport properties. However, unlike for SLC26A9, we did not detect specific currents that could be attributed to SLC26A6 in a wide voltage range. This was irrespective of whether we used symmetric $Cl^-$ solutions or asymmetric conditions, where the major anion in the extracellular solution was changed to $HCO_3^-$ (*Figure 1B*, *Figure 1—figure supplement 1B*). To confirm the expression of SLC26A6 at the plasma membrane, we employed surface biotinylation and found a similar level as observed for its electrogenic paralog SLC26A9 (*Figure 1C*). We thus concluded that the transporter would either be inactive, exceedingly slow or that transport would be electroneutral as a consequence of the strict 1:1 exchange of two monovalent anions.

Consequently, we turned to in vitro assays where we have reconstituted SLC26A6 into proteoliposomes and monitored its activity with ion-selective probes (*Figure 1—figure supplement 1C*). Initially, we investigated whether we would be able to detect any sign of uncoupled chloride flow by using a fluorometric assay based on the fluorophore ACMA. This assay monitors pH changes following the ionophore-mediated $H^+$ influx to compensate for the buildup of a membrane potential during $Cl^-$ uptake, which would enable us to observe transport even if the slow kinetics prevents characterization by electrophysiology (*Figure 1—figure supplement 1C*). In this way, we have previously monitored SLC26A9-mediated anion transport into proteoliposomes (*Walter et al., 2019*). However, in contrast to SLC26A9, no acidification was observed for SLC26A6, emphasizing that the transporter is either inactive or, alternatively, transport would be strictly coupled and electroneutral (*Figure 1D*).

Next, we were interested in whether $Cl^-$ would be exchanged by $HCO_3^-$ by a mechanism where the charges of transported anions are balanced. To this end, we directly monitored $Cl^-$ influx into $HCO_3^-$-loaded proteoliposomes using the fluorophore lucigenin, whose fluorescence is selectively quenched by the former anion (*Figure 1—figure supplement 1C*). In this assay, we found a robust time-dependent fluorescence decrease at high $Cl^-$ concentration, demonstrating its transport by SLC26A6 (*Figure 1E*, *Figure 1—figure supplement 1D, E*). The process did not require the addition of valinomycin to dissipate the membrane potential, thus underlining that transport is electroneutral. In the investigation of the $Cl^-$ concentration-dependence of transport, we found a saturation with an apparent $K_M$ of about 16–37 mM (*Figure 1F*, *Figure 1—figure supplement 1F*). This indicated low affinity for the transported anion is in the same range as observed for $Cl^-$ transport in SLC26A9 (*Walter et al., 2019*).

Finally, we probed whether $HCO_3^-$ itself would be a transported substrate. To this end, we employed a novel $HCO_3^-$-selective europium probe and incorporated this probe within proteoliposomes to assay $HCO_3^-$ influx (*Figure 1—figure supplement 1C*; *Martínez-Crespo et al., 2021*). The addition of 10 mM $HCO_3^-$ to the outside in presence of a symmetric 200 mM concentration of $Cl^-$ on both sides of the membrane caused a pronounced increase in luminescence that is well above the small non-specific signal observed in liposomes not containing any protein (*Figure 1G*). This increase is substantially enhanced when using a reduced outside $Cl^-$ concentration of 50 mM to generate an outwardly directed driving force (*Figure 1G*). Together these experiments demonstrate that $HCO_3^-$ is a transported anion and that its import into proteoliposomes is coupled to the counterflow of $Cl^-$. Taken together, our experiments underline the function of SLC26A6 as electroneutral coupled $Cl^-/HCO_3^-$ exchanger.

Besides $HCO_3^-$, SLC26A6 was also proposed to be involved in the transport of oxalate in exchange for $Cl^-$ and thus to participate in the excretion of this small divalent anion in both the intestine and kidneys (*Clark et al., 2008*; *Ohana et al., 2013*). However, if assuming a coupled 1:1 stoichiometry as observed in case of $Cl^-/HCO_3^-$ exchange, the transported charge would in this case no longer be

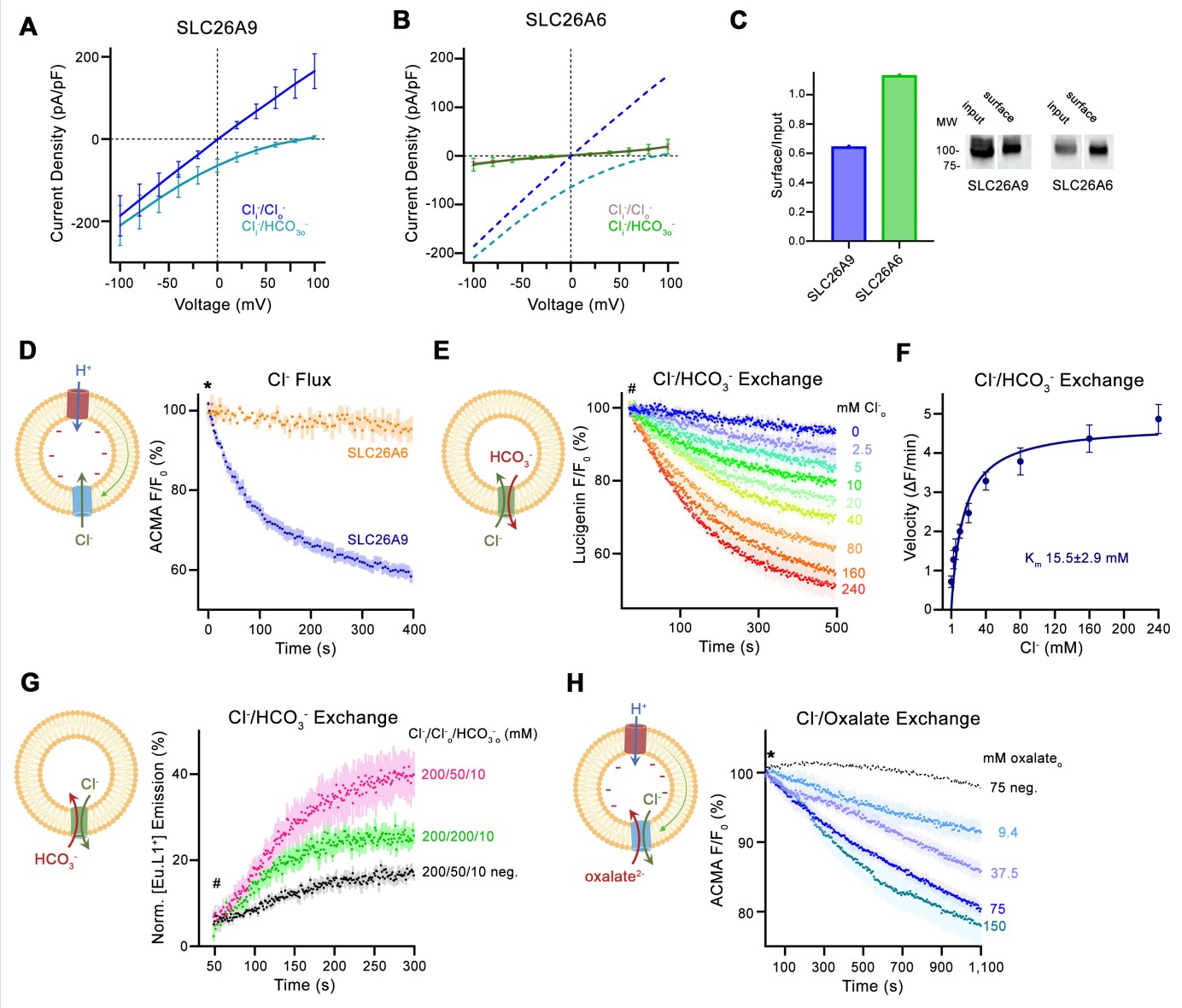

**Figure 1.** Transport properties of SLC26A6. (**A**) Current-Voltage relationships of HEK 293 cells expressing murine SLC26A9 and (**B**) human SLC26A6. Data were recorded in the whole-cell configuration in symmetric (150 mM) Cl⁻ concentrations and asymmetric conditions with equimolar (150 mM) concentrations of intracellular Cl⁻ and extracellular HCO₃⁻. Values show mean of independent experiments (SLC26A9 Cl⁻/Cl⁻ n=6, Cl⁻/ HCO₃⁻, n=6; SLC26A6 Cl⁻/Cl⁻ n=6, Cl⁻/ HCO₃⁻, n=6). Dashed lines in (**B**) correspond to SLC26A9 data displayed in (**A**). (**C**) Protein expression at the surface of HEK cells determined by surface biotinylation. Ratio of biotinylated (right) over total protein (left) as quantified from a Western blot against a myc-tag fused to the C-terminus of the respective constructs. (**D**) Uncoupled Cl⁻ transport mediated by either the modified murine construct SLC26A9ᵀ or SLC26A6 reconstituted into proteoliposomes, as monitored by the fluorescence change of the pH gradient-sensitive fluorophore ACMA. Traces show mean of seven and three replicates from two independent reconstitutions for SLC26A9ᵀ and SLC26A6. (**E**) Coupled Cl⁻/HCO₃⁻ exchange monitored by the time- and concentration-dependent quenching of the fluorophore lucigenin trapped inside proteoliposomes containing SLC26A6. Traces show mean of six independent experiments from two reconstitutions. (**F**) Cl⁻ concentration dependence of transport. Initial velocities were obtained from individual measurements displayed in (**E**), the solid line shows a fit to the Michaelis Menten equation with an apparent Kₘ of 16 mM. (**G**) Coupled Cl⁻/HCO₃⁻ exchange monitored by the time- and concentration-dependent luminescence increase of the HCO₃-selective probe [Eu.L1⁺] trapped inside proteoliposomes containing SLC26A6. Traces show mean of five independent experiments from three reconstitutions. 'neg.' refers to mock liposomes. (**E, G**), Hashtag indicates addition of the assayed anion. (**H**) Electrogenic oxalate uptake followed by the fluorescence change of the pH gradient sensitive fluorophore ACMA. Traces show mean quenching of ACMA fluorescence in a time- and concentration-dependent manner for SLC26A6 proteoliposomes with outside oxalate concentrations of 9.4 mM (n=3), 37.5 mM (n=5), 75 mM (n=6), 150 mM (n=8, all from two independent reconstitutions). Neg. refers to mock liposomes assayed upon addition of 75 mM oxalate as defined in ***Figure 1—figure supplement***

*Figure 1 continued on next page*

*Figure 1 continued*

*1G*. (**D, H**), Asterisk indicates addition of the H⁺ ionophore CCCP, which allows counterion movement and electrogenic Cl⁻ transport to proceed. (**A, B, D–H**), errors are s.e.m. (**D, E, G, H**) Scheme of the respective assay is shown left.

The online version of this article includes the following source data and figure supplement(s) for figure 1:

**Source data 1.** Electrophysiology, liposomal transport assay data and western blot.

**Figure supplement 1.** Transport data.

**Figure supplement 1—source data 1.** Extended electrophysiology and liposomal transport assay data.

balanced, thus converting an electroneutral into an electrogenic transport process. As there are no suitable fluorophores for the selective detection of oxalate and since the small dicarboxylic acid also interferes with the detection of Cl⁻ by lucigenin, we decided to investigate whether a potentially electrogenic Cl⁻/oxalate exchange could be assayed with the fluorophore ACMA (*Figure 1—figure supplement 1C*). In these experiments, we found a robust decrease of the ACMA fluorescence upon the addition of extracellular oxalate to Cl⁻-loaded proteoliposomes containing SLC26A6, with the concentration dependence and saturation of the fluorescence decay further emphasizing a specific transport process (*Figure 1H*). We also attempted to follow electrogenic Cl⁻/oxalate exchange by patch clamp electrophysiology in the whole cell configuration but did not measure pronounced currents that would be above background, likely due to the slow kinetics of the process (*Figure 1—figure supplement 1I*). Together, our functional experiments characterize SLC26A6 as a strictly coupled transporter that exchanges Cl⁻ with $HCO_3^-$ and presumably also oxalate at a 1:1 stoichiometric ratio.

## SLC26A6 structure

To reveal the molecular basis of the described transport properties of SLC26A6, we have determined its structure by cryo-electron microscopy (cryo-EM). A dataset of the protein purified in the detergent glycol-diosgenin (GDN) at 3.3 Å was of high quality and allowed for the unambiguous interpretation of the cryo-EM density by an atomic model (*Figure 2A*, *Figure 2—figure supplements 1 and 2*,*Table 1*). In its organization, the SLC26A6 structure shows familiar features that have previously been observed in various eukaryotic family members of known structure including human and murine SLC26A9 (*Chi et al., 2020*; *Walter et al., 2019*), SLC26A5 (*Bavi et al., 2021*; *Butan et al., 2022*; *Futamata et al., 2022*; *Ge et al., 2021*), SLC26A4 (*Liu et al., 2023*) and the plant transporter AtSULTR4 (*Wang et al., 2021*; *Figure 2B*, *Figure 2—figure supplement 3*). The SLC26A6 subunit consists of a membrane-inserted unit (TM) followed by a cytoplasmic STAS domain. In the homodimeric protein, subunit interactions are predominantly formed by the domain-swapped cytoplasmic parts including an extended N-terminal region and the C-terminal STAS domain, whereas the few contacts between the transmembrane transport units are confined to the end of the terminal membrane-spanning helix α14 (*Figure 2B*). A region connecting secondary structure elements of the STAS domain (located between Cα1 and β4) termed the intervening sequence (IVS) is present in the characterized protein (*Figure 2C*). Previously, removal of this region increased both the stability and surface expression of a construct of the murine SLC26A9 while retaining its transport properties (*Walter et al., 2019*). In the density, the N-terminal part of the IVS folds into an α-helix encompassing amino acids 569–593 (Cα$^{IVS}$) whereas the following 61 residues are unstructured and thus not resolved in the density (*Figure 2A*). On the C-terminus of the IVS, the density re-appears at Ser 655 at the region preceding β4, which interacts with Cα$^{IVS}$ (*Figure 2B and C*). A generally similar arrangement was also observed in a full-length construct of human SLC26A9 (*Chi et al., 2020*; *Figure 2D*). In both structures, Cα$^{IVS}$ runs about parallel to the membrane plane and contacts the loop connecting α8 and α9 of the mobile core domain. This presumably influences its conformational preferences thereby potentially impacting transport (*Figure 2B–D*).

## The transmembrane domain

In the dimeric transporter, the two TMs share minimal contacts and thus supposedly function as independent units (*Figure 2B*). Consequently, we expect structural differences underlying the distinct transport properties of SLC26 paralogs to be manifested in this part of the protein. Like in other pro- and eukaryotic family members of known structure, the TMs of SLC26A6 consist of two topologically related repeats of seven transmembrane segments, which span the membrane with opposite

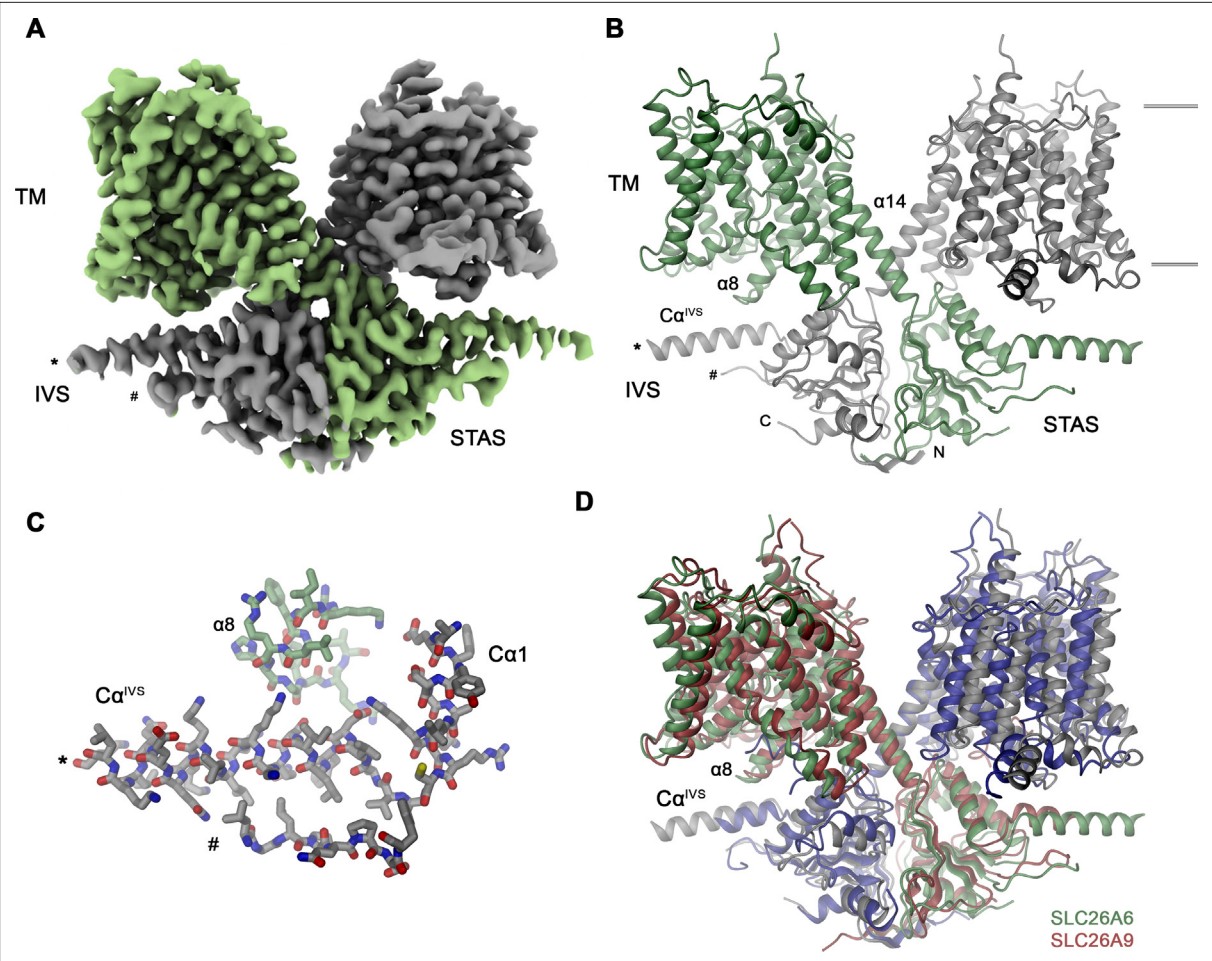

**Figure 2.** SLC26A6 structure. (**A**) Cryo-EM density of the SLC26A6 dimer at 3.28 Å and (**B**) ribbon representation of the model in the same orientation. Subunits are shown in unique colors and selected structural elements are labeled. The membrane boundary is indicated. (**C**) Interaction region between the loop proceeding α8 of the core domain and the helix Cα$^{IVS}$ of the adjacent subunit. (**A–C**) Start (*) and end (#) of the disordered region of the IVS are indicated. (**D**) Ribbon representation of the superimposed SLC26A6 (green, gray) and SLC26A9 (red, blue, PDBID: 7CH1) dimers.

The online version of this article includes the following figure supplement(s) for figure 2:

**Figure supplement 1.** Cryo-EM reconstruction of SLC26A6.

**Figure supplement 2.** Section of the cryo-EM density of SLC26A6.

**Figure supplement 3.** Sequence and Topology.

orientation (*Figure 2—figure supplement 3B*). Elements of both repeats are intertwined to form two segregated modules, a gate domain and a core domain (*Figure 3A*). The equivalence between the SLC26A6 and SLC26A9 conformations is illustrated in the overlay of the respective TMs whose Cα positions superimpose with an RMSD of 1.3 Å (*Figure 3B*). Both conformations represent inward-facing states of transporters presumably functioning by an alternate access mechanism. The substrate binding site located in the center of the core domain is accessible from the cytoplasm via a spacious aqueous cavity, whereas the extracellular exit is sealed by extended contacts between core and gate domains (*Figure 3C and D*, *Figure 3—figure supplement 1A, B*). The resemblance between individual sub-domains even exceeds the overall similarity of the TM (with core- and gate domains superimposing with RMSDs of 1.13 Å and 1.2 Å, respectively) distinguishing them as largely independent units of a modular protein (*Figure 3B*). In the comparison of both paralogs, the gate domains do not show pronounced differences (*Figure 3B*). Together with the cytoplasmic STAS domains, which constitute the bulk of the dimer interface, these sub-domains form a contiguous rigid scaffold of the dimeric protein that presumably remains static during transport (*Figure 2B*, *Figure 3—figure supplement 1C*). In contrast, both core domains represent the mobile elements of the protein (*Figure 3—figure*

**Table 1.** Cryo-EM data collection, refinement, and validation statistics.

| | hSLC26A6 in GDN (EMDB-17085) (PDB 8OPQ) | |
| --- | --- | --- |
| Data collection and processing | Data Set 1 | Data Set 2 |
| Magnification | 130,000 | 130,000 |
| Voltage (kV) | 300 | 300 |
| Electron exposure (e–/Å$^2$) | 61 | 67 |
| Defocus range (μm) | −0.8 to −2.4 | −1 to −2.4 |
| Pixel size (Å)* | 0.3255 (0.651) | 0.3255 (0.651) |
| Symmetry imposed | C2 | |
| Initial particle images (no.) | 1,749,907 | |
| Final particle images (no.) | 93,169 | |
| Map resolution (Å) FSC threshold 0.143 | 3.28 | |
| Map resolution range (Å) | 2.9–4.1 | |
| Refinement | | |
| Model resolution (Å) 0.5 FSC Threshold | 3.3 | |
| Model resolution range (Å) | 1.6–3.3 | |
| Map sharpening b-factor (Å$^2$) | 121.9 | |
| Model composition Non-hydrogen atoms Protein residues Ligands | 10,006 1296 2 | |
| *B* factors (Å$^2$) Protein Ligands | 58.64 30 | |
| R.m.s. deviations Bond lengths (Å) Bond angles (°) | 0.004 0.907 | |
| Validation MolProbity score Clashscore Poor rotamers (%) | 1.71 8.25 0 | |
| Ramachandran plot Favored (%) Allowed (%) Disallowed (%) | 96.18 3.82 0.0 | |

1. *Values in parentheses indicate the pixel size in super-resolution.

supplement 1D). Each core domain carries a substrate binding site located in a pocket at the center of the unit facing the gate domain (*Figures 3C, D , and 4A*). The presumable location of a bound chloride ion is resolved in the structure of SLC26A6 at a low contour of the map (*Figure 4B*, *Figure 2—figure supplement 2B*). This position concurs with positions inferred from known structures of paralogs, which show binding of $Cl^-$ and $HCO_3^-$ to an equivalent location (*Chi et al., 2020*; *Futamata et al., 2022*; *Ge et al., 2021*; *Liu et al., 2023*; *Figure 4B*, *Figure 4—figure supplement 1*). The anion binding site is placed between the opposed short helices α3 and α10, which partly span the membrane and whose N-termini are aligned to face each other in the center of the core domain (*Figure 4A*). Together with the side chains of surrounding residues, both α-helix termini form a selective anion binding site (*Figure 4B–G*, *Figure 4—figure supplement 1*). In complex with its bound cargo, the core domain is believed to shuttle as rigid unit between two extreme conformations where

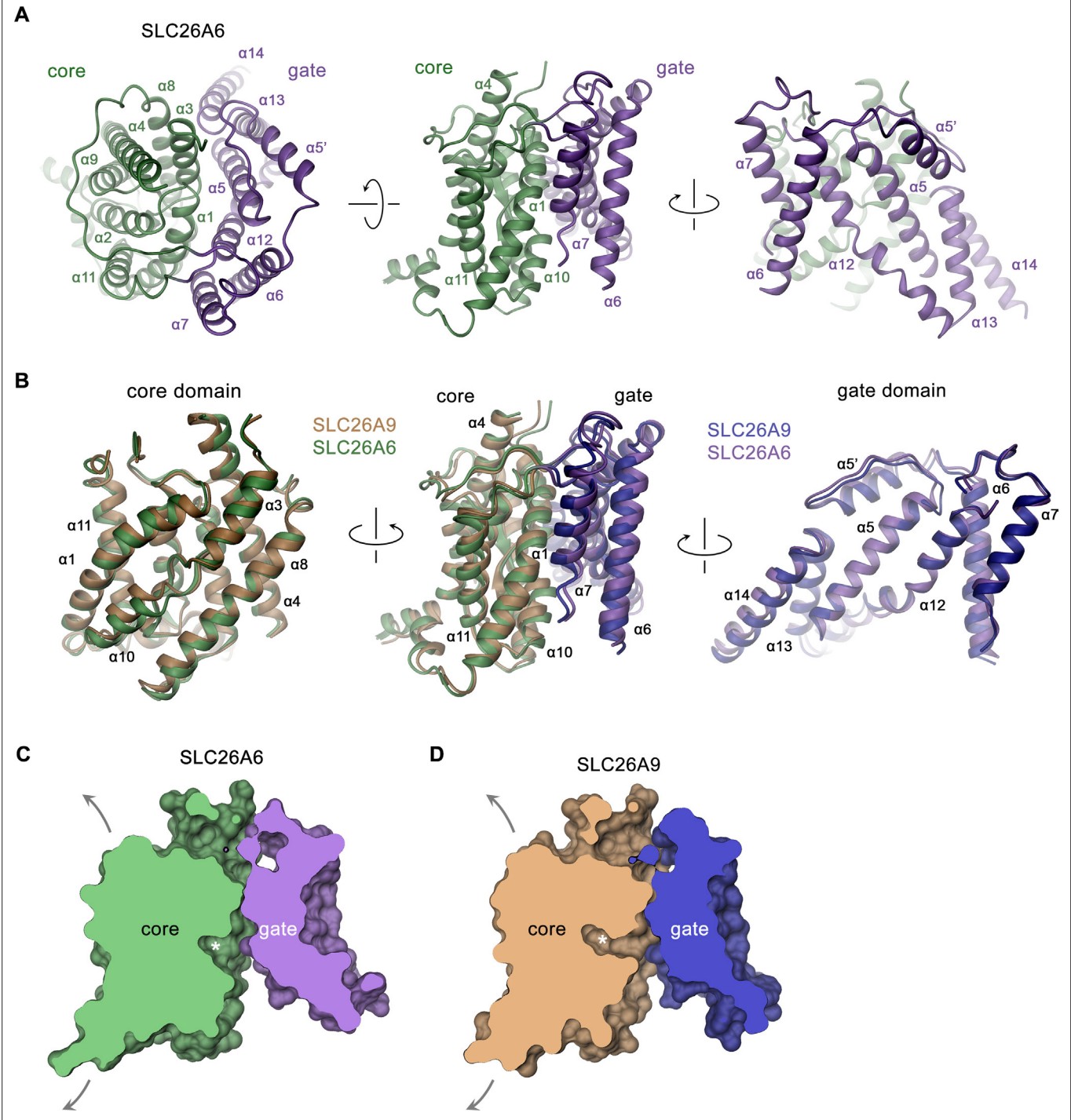

**Figure 3.** SLC26A6 TM domain. (**A**) Ribbon representation of the TM unit of SLC26A6 in indicated orientations (left, view is from the outside, center and right, from within the membrane). Core and gate domains are colored in green and violet, respectively. Selected secondary structure elements are labeled. (**B**) Superposition of elements of the TM between SLC26A6 and SLC26A9 (PDBID: 7CH1). Left, core domains, center, TMs, right, gate domains. Core and gate domains of SLC26A9 are colored in orange and blue, respectively. The view is from within the membrane with relative orientations indicated. (**C, D**) Slice across a surface of the TM domains of SLC26A6 (**C**) and SLC26A9 (**D**) viewed from within the membrane. The spacious aqueous cavity leading to the ion binding site from the cytoplasm is evident. Asterisk indicated the position of the transported ion. Arrows indicate possible movements of the core domain.

The online version of this article includes the following figure supplement(s) for figure 3:

**Figure supplement 1.** TM domain features.

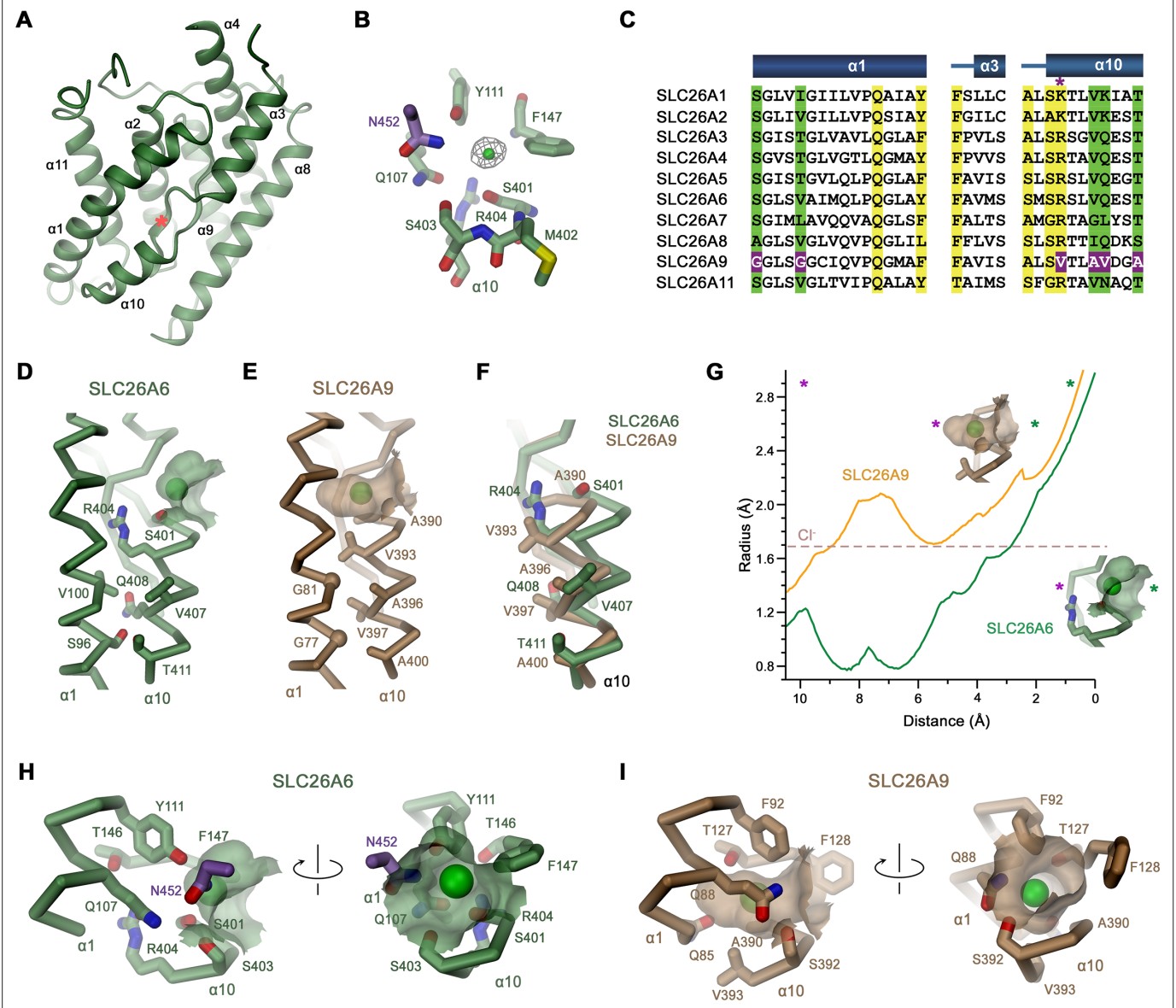

**Figure 4.** Features of the substrate binding site. (**A**) Ribbon representation of the core domain of SLC26A6 viewed from within the membrane from the gate domain. Asterisk indicates the location of the ion binding site, selected secondary structure elements are labeled. (**B**) Ion binding site with the density of a bound Cl⁻ ion (green) displayed as mesh. (**C**) Sequence alignment of the region constituting the ion binding site of the ten functional human SLC26 paralogs. Conserved residues in the contact region between α1 and α10 are highlighted in green, residues involved in ion interactions in yellow. Deviating residues in SLC26A9 are highlighted in violet. Asterisk marks position that harbors a basic residue in all family members except for SLC26A9, where the residue is replaced by a valine. Whereas most paralogs, including the ones operating as $HCO_3^-$ exchangers, carry an arginine at this site, the sulfate transporters SLC26A1 and 2 contain a smaller lysine. Secondary structure elements are shown above. (**D, E**) Cα-representation of the contact region between α1 and α10 of (**D**) SLC26A6 and (**E**) SLC26A9 (PDBID: 7CH1). (**F**) α10 of both transporters obtained from a superposition of the core domains. (**D–F**) Side chains of residues of the contact region and selected residues of the binding site are shown as sticks. (**G**) Size of the substrate cavity of SLC26A6 and SLC26A9 as calculated with HOLE (***Smart et al., 1996***). The radius of the substrate cavity of either protein is mapped along a trajectory connecting a start position at the entrance of each cavity (distance 0 Å) and an end position located outside of the cavity in the protein region (distance 10 Å). Both points are defined by asterisks in insets showing the substrate cavities for either transporter and they are indicated in the graph (green, cavity entrance towards the aqueous vestibule; violet, protein region). (**H, I**) Ion binding sites of SLC26A6 (**H**) and SLC26A9 (**I**). The relative orientation of views is indicated. (**D, E, H, I**) The position of bound ions was inferred as detailed in ***Figure 1***. The molecular surface surrounding the bound ions is displayed. Side chains of interacting residues are shown as sticks.

The online version of this article includes the following figure supplement(s) for figure 4:

**Figure supplement 1.** Binding site comparisons.

the binding site is either exposed to the cytoplasm or the extracellular environment (*Figure 3C and D*, *Figure 3—figure supplement 1D*).

As carrier of the substrate binding site, the core domains presumably display the characteristic features accounting for the distinct functional properties of SLC26A6 and SLC26A9. Their conservation within the SLC26 family is illustrated in the comparison of the respective domains of SLC26A6 and SLC26A5 (Prestin; *Figure 4C*, *Figure 4—figure supplement 1A, C, D, E, G*). Latter belongs to a protein that, instead of transporting ions, confers electromotility and which in the best defined structure (PDBID: 7LGU) resides in a distinct conformation where the substrate binding site has become buried between core and gate domain (leading to an RMSD of 2.9 Å for the entire TMD; *Ge et al., 2021*). Despite the functional and conformational differences, the core domains of SLC26A5 and SLC26A6 superimpose with an RMSD of only 1.13 Å, illustrating their close structural resemblance (*Figure 4—figure supplement 1A, C*). Both proteins share similar ion binding properties that extends to Cl⁻ and $HCO_3^-$, latter of which is not a substrate of SLC26A9 (*Walter et al., 2019*). In contrast to the general similarity with SLC26A5, the comparison of the same units of SLC26A6 and SLC26A9 shows distinct features that likely underlie the differences in substrate selectivity and transport mechanism (*Figure 4—figure supplement 1B,C*). These are confined to the conformation of α10, which at its N-terminus forms direct interactions with transported ions, whereas the remainder of the domain is very similar (*Figure 4—figure supplement 1B,C*). The difference in the conformations of this short membrane-inserted helix can be approximated by a 12–14° rotation of α10 of SLC26A9 around an axis located at its C-terminal end towards α-helix 1 (*Figure 4F*, *Figure 4—figure supplement 1B,C*). The replacement of residues within the contact region of SLC26A9 in comparison to other family members acts to lower their side chain volume at the contact region between both helices (*Figure 4C–F*). These include Ser 96 and Val 100 on SLC26A6, which are both replaced by glycines at equivalent positions in SLC26A9 (i.e., Gly 77 and Gly 81) and a substitution of Val 407, Gln 408 and Thr 411 by Ala 396, Val 397 and Ala 400 (*Figure 4C–F*). In addition to the described substitutions, there are also differences in the side chains surrounding the ion binding site, which together determine the size and polarity of the substrate binding pocket (*Figure 4C*). These differences extend to Ser 401 of SLC26A6, which is replaced by an alanine in SLC26A9 (Ala 390), and most prominent, the large Arg 404 of SLC26A6, which is replaced by Val 393 in SLC26A9 (*Figure 4C-I*, *Figure 4—figure supplement 1D–G*). As a result, the pocket is considerably deeper in SLC26A9 compared SLC26A6, where it is delimited by the side chain of Arg 404 that interacts with the residues preceding α helix 3 to form the back side of the shallow binding site (*Figure 4D–I*, *Figure 4—figure supplement 1D–H*). In addition to its structural role, Arg 404 likely also contributes to the stabilization of bound anions by coulombic interactions, with its guanidium group potentially engaging in direct interactions with bicarbonate or oxalate. The altered geometry of the site leads to the binding of Cl⁻ deeper in the pocket in case of SLC26A9 (*Figure 4D, E,G-I*, *Figure 4—figure supplement 1H*) and presumably explains why bicarbonate interacts with SLC26A6 but not SLC26A9.

## Structure-function relationships

Whereas our functional data has defined SLC26A6 as a coupled antiporter that exchanges Cl⁻ with $HCO_3^-$ and presumably also oxalate with equimolar stoichiometry, its structure has revealed the architecture of a transport protein in an inward-facing conformation. The transport properties are presumably determined by the detailed organization of the mobile core domain. This protein module shows pronounced structural differences compared to the equivalent unit of the uncoupled transporter SLC26A9, resulting in a remodeled anion binding site, which in SLC26A6 is lined by a conserved basic residue (i.e. Arg 404) that is shared by most but not all mammalian paralogs (*Figure 4C*, *Figure 4—figure supplement 1G*, *Figure 5A*). We thus decided to mutate this position to the amino acid found in SLC26A9 and to characterize the transport properties of the SLC26A6 mutant R404V. This mutation is well-tolerated and does not interfere with protein integrity as judged by its expression level and biochemical properties. We initially investigated whether SLC26A6 R404V would conduct ions as a consequence of compromised coupling properties and thus studied the protein by patch-clamp electrophysiology. In whole-cell patch clamp experiments, we detected small currents in symmetric Cl⁻ solutions, which are low compared to SLC26A9, despite the comparable expression of the mutant at the cell-surface (*Figure 5B–E*). Irrespective of their magnitude, which is not significantly higher than SLC26A6 WT (*Figure 5C*

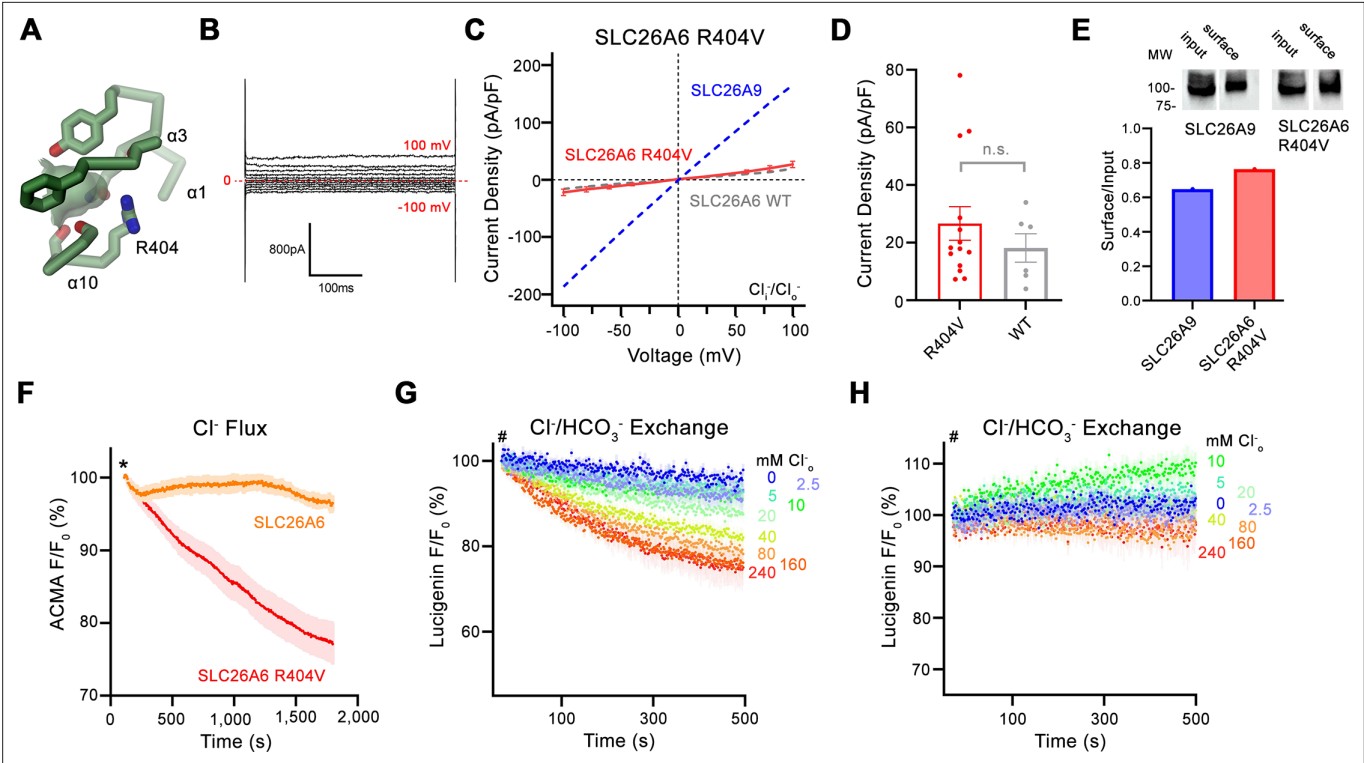

**Figure 5.** Functional properties of a structure based SLC26A6 construct. (**A**) SLC26 ion binding site showing surrounding residues including Arg 404. (**B**) Representative current trace and (**C**) current-voltage relationships of HEK 293 cells expressing the SLC26A6 mutant R404V. Data were recorded in the whole-cell configuration at symmetric (150 mM) Cl⁻ concentrations. Values show mean of 14 independent experiments. Dashed lines correspond to SLC26A9 (blue) and SLC26A6 data (grey) displayed in *Figure 1A and B*. (**D**) Average current densities of the SLC26A6 mutant R404V (n=14) and SLC26A6 WT (n=6). Values show currents recorded at 100 mV as displayed in **C** and *Figure 1B*. Although the currents were consistently larger for the mutant R404V, the difference is not statistically significant (p=0.69). (**E**) Protein expression of SLC26A6 R404V and SLC26A9 at the surface of HEK cells determined by surface biotinylation. Ratio of biotinylated (right) over total protein (left) as quantified from a Western blot against myc-tag fused to the C-terminus of the respective constructs. (**F**) Uncoupled Cl⁻ transport mediated by SLC26A6 R404V reconstituted into proteoliposomes, as monitored by the fluorescence change of the pH gradient-sensitive fluorophore ACMA. Traces of SLC26A6 are shown for comparison. Data shows mean of five replicates from two independent reconstitutions for both constructs. (**G, H**) Coupled Cl⁻/HCO₃⁻ exchange by the SLC26A6 mutant R404V monitored by the time- and concentration- dependent quenching of the fluorophore lucigenin trapped inside proteoliposomes. (**G**) Uncorrected traces and (**H**) traces corrected by the background obtained from empty liposomes displayed in *Figure 1—figure supplement 1D*, which do not show indication of transport. (**G, H**) Traces show mean of five independent experiments from two reconstitutions. (**C, D, F, G, H**) errors are s.e.m.

The online version of this article includes the following source data for figure 5:

**Source data 1.** Transport data and Western blot of the SLC26A6 mutant R404V.

*and D*), these currents are indicative for the capability of the mutant to mediate electrogenic Cl⁻ transport at low rates. We thus turned to our proteoliposome-based ACMA assay to confirm that side chain replacement has indeed conferred the ability to SLC26A6 to pass downhill Cl⁻ transport (*Figure 5F*). After demonstrating uncoupled chloride transport, we tested whether the mutant has retained its ability of mediating coupled Cl⁻/HCO₃⁻ exchange using the fluorophore lucigenin but did not find pronounced Cl⁻ flux in this case (*Figure 5G and H*). Together our data suggest that the mutation of a conserved basic position in the ion binding site has compromised the ability of SLC26A6 to mediate coupled Cl⁻/HCO₃⁻ exchange and instead facilitated uncoupled Cl⁻ transport with slow kinetics. While these findings underline the importance of Arg 404 for anion interactions and coupling, they also emphasize that the mutation of a single residue is insufficient to confer the complete functional phenotype of SLC26A9 as a fast uncoupled Cl⁻ conductor. Nevertheless, our results further underline the importance of the anion binding site in SLC26 transporters for the transport mechanism of its members.

## Discussion

By combining structural data obtained by cryo-EM with electrophysiology and transport assays, our study has elucidated the previously unknown architecture of SLC26A6, defined its transport properties and revealed the features that underlie the mechanistic distinction to the paralog SLC26A9. By using a novel $HCO_3^-$ selective europium probe (*Martínez-Crespo et al., 2021*), we have directly demonstrated the transport of this important physiological anion (*Figure 1G*). We have further clarified the controversy concerning the stoichiometry of anions transported by SLC26A6 (*Ohana et al., 2009*). In the case of $Cl^-/HCO_3^-$ transport, we detect a strict equimolar exchange of anions binding to a conserved site in the mobile core domain of the transmembrane transport unit (*Figure 4B and H*). Although not shown unambiguously, we assume an analogous mechanism also for $Cl^-$/oxalate exchange. Consequently, transport would be electroneutral in case of the monovalent $HCO_3^-$ and electrogenic in case of the divalent oxalate (*Figure 1E–H*), which was already proposed in a previous study (*Chernova et al., 2005*).

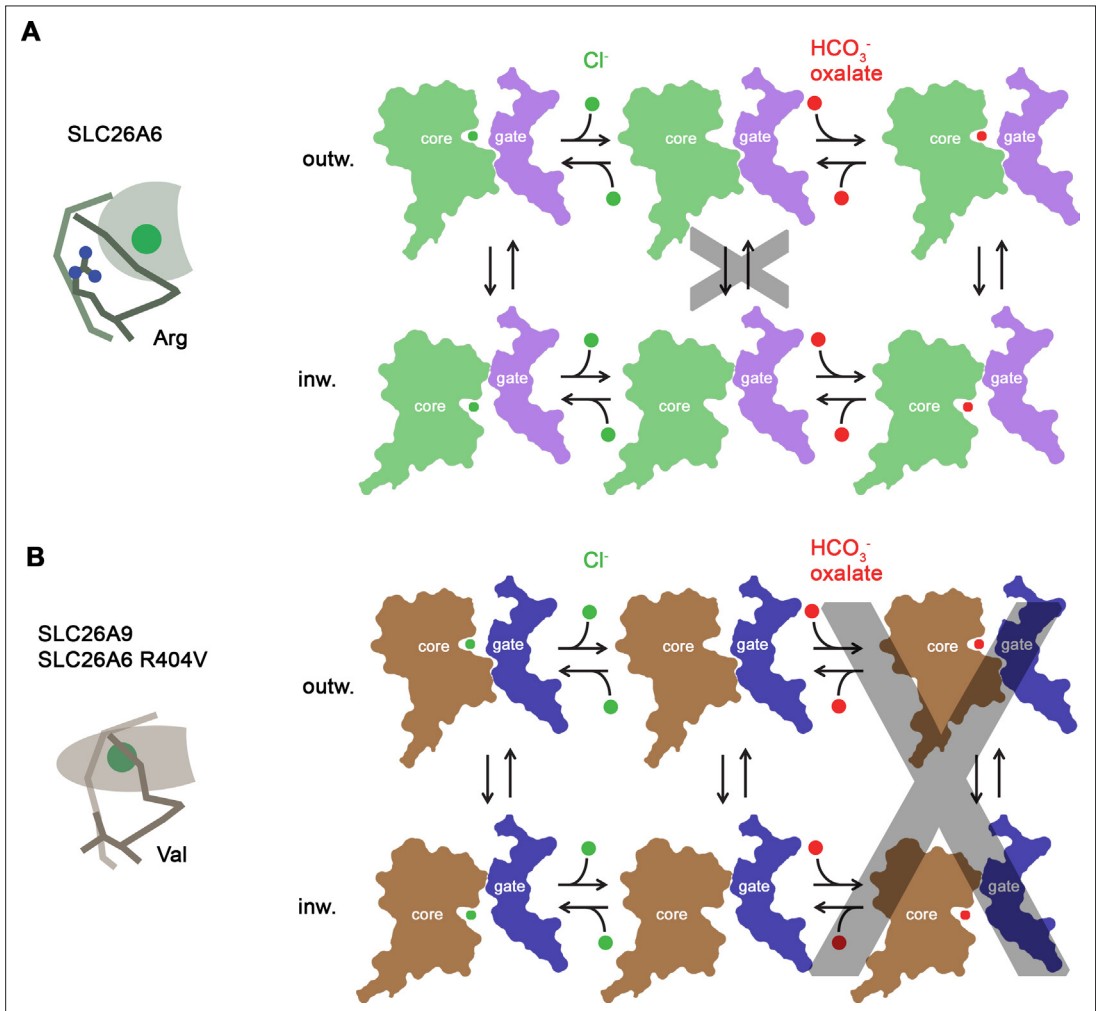

**Figure 6.** Transport mechanism. Features of the anion binding site (left) and kinetic schemes (right) of two SLC26 paralogs with distinct functional properties. (**A**) The coupled antiporter SLC26A6 mediates the strict stoichiometric exchange of $Cl^-$ and $HCO_3^-$ and presumably also of oxalate. The protein readily cycles between inward- and outward-facing conformations in substrate-loaded states, whereas the transition in an unloaded state is kinetically disfavored. The binding of different anions is facilitated by a large but shallow binding site with high field-strength. (**B**) The uncoupled $Cl^-$ transporter SLC26A9 has a narrower substrate selectivity where both oxalate and $HCO_3^-$ are not among the transported ions. Uncoupled $Cl^-$ transport is likely mediated by a mechanism that allows the rapid transition of the unloaded transporter between inward- and outward-facing conformations. The transported ion binds to a site with low field-strength. A similar mechanism, although with slower kinetics, is mediated by a point mutant of SLC26A6 where a conserved Arg of the binding site is replaced by a Val, the corresponding residue found in SLC26A9.

Regarding their function, SLC26A6 and SLC26A9 show different features with respect to substrate selectivity and coupling (*Ohana et al., 2009*). Remarkably, both proteins are capable of interacting with Cl$^-$ with similar mM affinity (*Figure 1F*; *Walter et al., 2019*). However, they do this via distinct interactions in equivalent binding regions. In SLC26A9, Cl$^-$ binds in a deep pocket with low field strength, whereas the same pocket is shallower in SLC26A6, where a conserved arginine (Arg 404) increases the positive electrostatic potential and thus likely stabilizes the bound anion (*Figure 4H, I* and *Figure 6*). Together with the presence of additional residues at the binding site (i.e., Ser 401 at the N-terminal end of α10), the captured Cl$^-$ in SLC26A6 thus can rely on a larger density of polar interactions (*Figure 4—figure supplement 1E,F*). The same site likely also provides a suitable environment for HCO$_3^-$ and oxalate, neither of which are substrates of SLC26A9 (*Figure 1A*, *Figure 1—figure supplement 1A,H*; *Dorwart et al., 2007*; *Walter et al., 2019*). A similar anion interaction observed in SLC26A6 is also found in the paralogs SLC26A4 (Pendrin) (*Liu et al., 2023*) and SLC26A5 (Prestin) (*Futamata et al., 2022*; *Ge et al., 2021*), which both share a similar substrate preference. The similarity between the core domains of SLC26A6 and Prestin is remarkable in light of the altered function of the latter, which is no longer capable of transporting ions and instead operates as motor protein in cochlear outer hair cells of mammals (*Zheng et al., 2000*). While to some degree Prestin undergoes comparable conformational changes as other family members, as illustrated in the close relationship between occluded conformations of SLC26A9 and SLC26A5, the full transition into an outward-facing conformation to release the bound anion to the extracellular environment appears to be prohibited (*Bavi et al., 2021*; *Ge et al., 2021*).

Besides the absence of a residue carrying a positive charge, the altered binding site in SLC26A9 is also a consequence of concerted replacements of interacting amino acids on α1 and α10, which change the immediate environment of the bound ion and allow for a reorientation of the partly inserted helix α10 whose N-terminus contributes part of the interactions (*Figure 4C and D*). Collectively, these changes presumably lead to the altered ion selectivity. The importance of the described anion binding site for the transport mechanism is illustrated in the point mutation R404V in SLC26A6. The replacement of this basic residue, which was considered a hallmark in the distinction between SLC26A9 and other family members (*Walter et al., 2019*), has compromised coupled Cl$^-$/HCO$_3^-$ exchange and instead mediates uncoupled Cl$^-$ transport, although with considerably slower kinetics than in SLC26A9 (*Figures 5B–G and 6*).

From a mechanistic viewpoint SLC26A9 and SLC26A6 are distinguished by the sequence of conformations that together define the transport cycle (*Figure 6*). In the uncoupled SLC26A9, the transporter is able to transition between inward- and outward facing states at a high rate, which underlies the pronounced currents observed in patch clamp experiments (*Figure 6B*, *Figure 1—figure supplement 1A*). The same change of the access in an unloaded conformation is kinetically unfavorable in the coupled exchanger SLC26A6, as illustrated in the lack of current in electrophysiological recordings and the absence of a Cl$^-$ leak in a vesicle-based assay (*Figures 1B–D , and 6A*, *Figure 1—figure supplement 1B*). The basis for the distinctive kinetic features of both paralogs is currently still unclear, and we do at this stage not exclude the existence of a leaky channel-like conformation in SLC26A9. Such a conformation could potentially promote Cl$^-$ flow across a continuous aqueous pore without obligatory protein movement, although there is currently no structural evidence for its existence. The nature of the energy barrier, which prevents the shuttling of the unloaded state of SLC26A6 and other paralogs with similar functional phenotype (i.e. the transporters SLC26A1-4) has thus far remained elusive. It is likely that conformational changes are dictated by interactions between the mobile core domain with the immobile gate domain, which are both believed to operate as semi-independent entities (*Geertsma et al., 2015*; *Walter et al., 2019*). Remarkably, both units bury a similar contact area in the inward-facing conformations of SLC26A6 and SLC26A9 and do not exhibit pronounced differences in their interactions (*Figure 3—figure supplement 1A,B*). Additional contacts in SLC26A6 are established with the structured N-terminal part of the IVS region of the STAS domain of the interacting subunit, which appear to further stabilize the observed inward-facing conformation (*Figure 2C*). Similar interactions have also been found in the full-length transporter SLC26A9 (*Chi et al., 2020*), which, next to the increased surface expression, could account for the strongly increased activity observed in the construct SLC26A9$^T$ where the IVS sequence of the protein was removed (*Walter et al., 2019*). Besides the effect of the IVS, also the truncation of the C-terminus of SLC26A9 might have contributed to this high activity. The unstructured C-termini of family

members are weakly conserved and in case of SLC26A6 and SLC26A9, they contain a PDZ binding motif at their end, which is believed to tether either transporter to potential partners such as the anion channel CFTR via interacting scaffolding proteins (*Shcheynikov et al., 2006a*). In the isoform b of SLC26A9, a C-terminal extension was located in the aqueous cavity leading to the substrate binding site in the inward-facing conformation of SLC26A9, thereby locking the transporter in the observed conformation (*Chi et al., 2020*; *Figure 2—figure supplement 3A*). There is no similar extension found in any SLC26A6 isoforms, and the role of this interaction in a cellular environment is currently unclear, although it hints at a possible regulation in SLC26A9. Collectively, our study has provided novel insight into the structural basis of substrate selectivity and the mechanistic distinction between uncoupled and coupled transporters of the SLC26 family and thus provides a foundation for future investigations.

# Materials and methods

**Key resources table**

| Reagent type (species) or resource | Designation | Source or reference | Identifiers | Additional information |
|---|---|---|---|---|
| Chemical compound, drug | HyClone HyCell TransFx-H medium | Cytiva | SH30939.02 | |
| Chemical compound, drug | 1-palmitoyl-2-oleoyl-sn-glycero-3-phosphoethanolamine (POPE) | Avanti Polar Lipids | 850757 C | |
| Chemical compound, drug | 1-palmitoyl-2-oleoyl-sn-glycero-3-phospho-(1'-rac-glycerol) (POPG) | Avanti Polar Lipids | 840457 C | |
| Chemical compound, drug | 1-palmitoyl-2-oleoyl-glycero-3-phosphocholine (POPC) | Avanti Polar Lipids | 850457 C | |
| Chemical compound, drug | Cholesterol | Sigma | C8667 | |
| Chemical compound, drug | Triton X-100 | Sigma | T9284 | |
| Chemical compound, drug | Pepstatin A | Axon lab | A2205.0100 | |
| Chemical compound, drug | Phenylmethylsulfonyl fluoride (PMSF) | Sigma | PMSF-RO | |
| Chemical compound, drug | Leupeptin | AppliChem | A2183,0100 | |
| Chemical compound, drug | Benzamidine | Sigma | B6506 | |
| Chemical compound, drug | Chloroform | Fluka | 25690 | |
| Chemical compound, drug | Glyco-diosgenin (GDN) | Anatrace | GDN101 | |
| Chemical compound, drug | Diethyl ether | Sigma | 296082 | |
| Chemical compound, drug | DNase I | AppliChem | A3778 | |
| Chemical compound, drug | Glycerin, Glycerol 86% | Roth | 4043.3 | |
| Chemical compound, drug | HCl | Merck Millipore | 1.00319.1000 | |
| Chemical compound, drug | HEPES | Sigma | H3375 | |
| Chemical compound, drug | [Eu.L1]$^+$ | Loughborough University (Dr SJ Butler) | N/A | |

*Continued on next page*

*Continued*

| Reagent type (species) or resource | Designation | Source or reference | Identifiers | Additional information |
|---|---|---|---|---|
| Chemical compound, drug | ACMA | Thermofischer Scientific | A1324 | |
| Chemical compound, drug | CCCP | Sigma | C2759 | |
| Chemical compound, drug | *N,N'*-Dimethyl-9,9'-biacridinium-dinitrat (Lucigenin) | Sigma | M8010 | |
| Chemical compound, drug | Phosphate buffered saline | Sigma | D8537 | |
| Chemical compound, drug | Strep-Tactin Superflow high capacity resin | IBA | 2-1208-010 | |
| Chemical compound, drug | D-desthiobiotin | Sigma | D1411 | |
| Chemical compound, drug | Kolliphor P188 | Sigma | K4894 | |
| Chemical compound, drug | L-glutamine | Sigma | G7513 | |
| Chemical compound, drug | Penicillin-streptomycin | Sigma | P0781 | |
| Chemical compound, drug | Fetal bovine serum | Sigma | F7524 | |
| Chemical compound, drug | Polyethylenimine 25 K MW, linear | Polysciences | 23966–1 | |
| Chemical compound, drug | 40 kDa linear PEI MAX | Polysciences | 24765–1 | |
| Chemical compound, drug | Valproic acid | Sigma | P4543 | |
| Chemical compound, drug | Calcium Chloride | Sigma | 223506 | |
| Chemical compound, drug | Magnesium Chloride | Fluka | 63,065 | |
| Chemical compound, drug | Potassium chloride | Sigma | 746346 | |
| Chemical compound, drug | Sodium chloride | Sigma | 71380 | |
| Chemical compound, drug | Terrific broth | Sigma | T9179 | |
| Chemical compound, drug | Mouse-anti-myc primary antibody | Sigma | M4439 | (WB 1:5000) |
| Chemical compound, drug | Peroxidase AffiniPure polyclonal Goat Anti-Mouse IgG (H+L) | Jackson ImmunoResearch | 115-035-003 RRID: AB_10015289 | (WB 1:10000) |
| Commercial assay or kit | 4–20% Mini-PROTEAN TGX Precast Protein Gels, 15-well, 15 µl | BioRad Laboratories | 4561096DC | |
| Commercial assay or kit | Amicon Ultra-4 Centrifugal Filters Ultracel 100 K, 4 ml | Merck Millipore | UFC810024 | |
| Commercial assay or kit | Borosilicate glass capillary with filament | Sutter Instrument | BF150-86-10HP | |
| Commercial assay or kit | 0.22 µm Ultrafree-MC Centrifugal Filtfer | EMD Millipore | UFC30GV | |

*Continued on next page*

*Continued*

| Reagent type (species) or resource | Designation | Source or reference | Identifiers | Additional information |
|---|---|---|---|---|
| Commercial assay or kit | Biobeads SM-2 adsorbents | BioRad Laboratories | 152–3920 | |
| Commercial assay or kit | Avestin Extruder kit | Sigma | Z373400 | |
| Commercial assay or kit | Pierce Cell Surface Biotinylation and Isolation Kit | Thermofischer Scientific | A44390 | |
| Commercial assay or kit | Amersham ECL Prime Western Blotting Detection Reagent | Cytiva | RPN2232 | |
| Commercial assay or kit | Polycarbonate Membranes 400 nm | Sigma | 610007 | |
| Commercial assay or kit | Polycarbonate Membranes 50 nm | Sigma | 610003 | |
| Commercial assay or kit | 96-well black walled microplates | Thermofischer Scientific | M33089 | |
| Commercial assay or kit | 384-well black microplate flat-bottom | Greiner | 781076 | |
| Commercial assay or kit | Quantifoil R1.2/1.3 Au 200 Mesh | Electron Microscopy Sciences | Q2100AR1.3 | |
| Commercial assay or kit | Superose 6 10/300 GL | Cytiva | 17517501 | |
| Other | BioQuantum Energy Filter | Gatan | N/A | |
| Other | HPL6 | Maximator | N/A | |
| Other | K3 Direct Detector | Gatan | N/A | |
| Other | Titan Krios G3i | ThermoFisher Scientific | N/A | |
| Other | Viber Fusion FX7 imaging system | Witec | N/A | |
| Other | TECAN M1000 Infinite | TECAN | N/A | |
| Other | TECAN SPARK | TECAN | N/A | |
| Other | Vitrobot Mark IV | ThermoFisher Scientific | N/A | |
| Cell line (human) | HEK293S GnTi- & HEK293T | ATCC | CRL-3022 & CRL-1573 | |
| Recombinant DNA reagent | Mammalian expression vector with C-terminal 3 C cleavage site, venus fluorescent tag, myc tag, SBP tag. | Dutzler laboratory | N/A | |
| Recombinant DNA reagent | *Mus musculus* SLC26A9 ORF shuttle clone | BioScience | GenBank BC160193 | |
| Recombinant DNA reagent | *Homo sapiens* SLC26A6 cDNA clone | BioScience | GenBank BC017697 | |
| Recombinant protein | HRV 3 C protease | Expressed (pET_3 C) and purified in Dutzler laboratory | N/A | |
| Software, algorithm | ASTRA7.2 | Wyatt Technology | RRID:SCR_016255 https://www.wyatt.com/products/software/astra.html | |
| Software, algorithm | ChimeraX 1.4 | *Pettersen et al., 2021* | RRID:SCR_015872 https://www.rbvi.ucsf.edu/chimerax/ | |

*Continued on next page*

*Continued*

| Reagent type (species) or resource | Designation | Source or reference | Identifiers | Additional information |
|---|---|---|---|---|
| Software, algorithm | Biorender | https://app.biorender.com/biorender-templates | | |
| Software, algorithm | Coot v.0.9.4 | *Emsley et al., 2010* | RRID:SCR_014222 https://www2.mrc-lmb.cam.ac.uk/personal/pemsley/coot/ | |
| Software, algorithm | cryoSPARC v3.2.0–4.0 | Structura Biotechnology Inc. | RRID:SCR_016501 https://cryosparc.com/ | |
| Software, algorithm | DINO | | RRID:SCR_013497 http://www.dino3d.org | |
| Software, algorithm | EPU2.9 | Thermo Fisher Scientific | N/A | |
| Software, algorithm | Phenix | *Liebschner et al., 2019* | RRID:SCR_014224 https://www.phenix-online.org/ | |
| Software, algorithm | DeepEMhancer | *Sanchez-Garcia et al., 2021* | https://github.com/rsanchezgarc/deepEMhancer | |
| Software, algorithm | JALVIEW | *Waterhouse et al., 2009* | | |
| Software, algorithm | Muscle | Edgar, 2004 | | |
| Software, algorithm | Axon Clampex 10.6 | Molecular Devices | N/A | |
| Software, algorithm | Axon Clampfit 10.6 | Molecular Devices | N/A | |
| Software, algorithm | Prism 9 | GraphPad | N/A | |
| Strain, strain background (E Coli) | *E. coli* MC1061 | Thermo Fisher Scientific | C66303 | |

## Construct generation

The open reading frame of human SLC26A6 (isoform 3, GenBank accession number: BC017697) and mouse SLC26A9 (GenBank accession number: BC160193) was amplified from a cDNA clone (Source BioScience) and cloned into into a pcDNA 3.1 vector (Invitrogen) modified to be compatible with the FX-cloning (*Geertsma and Dutzler, 2011*). The truncated murine SLC26A9 construct (SLC26A9$^T$) was generated as previously detailed (*Walter et al., 2019*). All FX-modified expression constructs contained a C-terminal C3 protease cleavage site, venus YFP, myc-tag and streptavidin-binding protein (SBP). As a consequence of FX cloning, expressed constructs include an additional serine at the N-terminus and an alanine at the C-terminus. Following C3-cleavage, SLC26A6 carries a seven residues long C-terminal extension (of sequence ALEVLFQ). For the generation of the hSLC26A6 point mutant R404V, the QuickChange (Aligent) mutagenesis method was used with the following primers: Forward: 5'-GCT CTA TGT CTG TGA GCC TGG TAC-3' and Reverse: 5'-GTA CCA GGC TCA CAG ACA TAG AGC-3'.

## Protein expression

Both HEK293S GnTI$^-$ (ATCC CRL-3022) and HEK293T (ATCC-1573) cell-lines used for protein expression tested negative for mycoplasma contamination.

Suspension HEK HEK293S GnTI$^-$ cells were grown and maintained in HyClone TransFx-H (GE Healthcare) media supplemented with 1% FBS, 1% Kolliphor P188, 100 U ml$^{-1}$ penicillin/streptomycin and 2 mM Glutamine. Cells were incubated at 37 °C and 5% CO$_2$ in TubeSpin Bioreactor 600 vessels (TPP) and shaken at 185 rpm.

For transient transfections, cells were split for transfection 24 hr prior from a cell density of 1–1.5 10$^6$ ml$^{-1}$–0.4-0.6 10$^6$ ml$^{-1}$. A ratio of 1.2 µg Plasmid DNA per 10$^6$ cells and 40 kDa linear PEI MAX (Polysciences) transfection reagent was used for protein expression. On the day of transfection, a 10 µg ml$^{-1}$ DNA dilution was incubated in non-supplemented DMEM media (Sigma). Linear PEI MAX was incubated with DNA at a ratio of 1:3 (w/w) from a 1 mg ml$^{-1}$ stock. After a 15 min incubation, the

DNA-PEI mix was added to suspension cells and co-incubated with 3 mM valproic acid (VPA). Cells were harvested after 48 hr and flash-frozen in liquid $N_2$ prior to storage at –80 °C.

## Protein purification

For protein purification, all protocols were performed at 4 °C. Cell pellets were thawed and mixed with extraction buffer (25 mM HEPES, pH 7.4, 200 mM NaCl, 5% glycerol, 2 mM $CaCl_2$, 2 mM $MgCl_2$, 2% glycol-diosgenin (GDN), 1 mM benzamidine, 10 µM leupeptin, 1 µM pepstatin A, 100 µM PMSF). For lysis cells were incubated for 2 hr at 4 °C under gentle agitation prior to ultracentrifugation (85,0000 $g$, 30 min, 4 °C) to remove insoluble material. Lysate was filtered (5 µm filter, Sartorius) prior to binding to Streptactin Superflow high-capacity resin (IBA Lifesciences) equilibrated in wash buffer (25 mM HEPES, pH 7.4, 200 mM NaCl, 5% glycerol, 0.02% GDN) for 2 hr.

After discarding the flow-through of clarified lysate through gravity flow, Streptactin resin was washed with 40 column volumes (CV) wash buffer and SBP-tagged protein was eluted using wash buffer supplemented with 10 mM desthiobiotin (Sigma). Eluate was concentrated to 500 µl using a 100 kDa MWCO concentrator (amicon) pre-equilibrated in wash buffer. C-terminal purification and detection tags were cleaved by 3 C protease at a 2:1 protease:protein mass ratio. Prior to size-exclusion chromatography, samples were passed through a 0.22 µm filter (Millipore). Samples were injected on the Äkta prime plus chromatography system (Cytivia) and separated on a Superose 6 10/300 column (GE Healthcare) in SEC buffer (10 mM HEPES, pH 7.4, 200 mM NaCl, 0.02% GDN). Protein containing peak samples were collected and used for cryo-EM sample preparation at a 2 mg ml$^{-1}$ concentration or used immediately for liposome reconstitution.

## Surface biotinylation

Adherent HEK293T cells were transfected at 80% confluency were transfected with 10 µg of plasmid encoding SLC26A6 WT and R404V constructs fused with a C-terminal SBP-Myc-Venus tag per 10 cm culture dish.

Transfection was performed using PEI MAX at a DNA:PEI ratio of 1:4. The Pierce Cell Surface Biotinylation and Isolation Kit (Themo Fisher) was used according to manufacturer's guidelines. Cells were washed with PBS from 10 ml culture per construct and biotinylated following 24 hr of expression using EZ-Link Sulfo-NHS-SS-Biotin. Quenching was then performed using 750 µl quenching buffer and cells were harvested. Following a PBS wash-step and centrifugation at 1000 $g$, cells were re-suspended and lysed in 200 µl extraction buffer (25 mM HEPES, pH 7.4, 200 mM NaCl, 5% glycerol, 2 mM $CaCl_2$, 2 mM $MgCl_2$, 2% GDN, 1 mM benzamidine, 10 µM leupeptin, 1 µM pepstatin A, 100 µM PMSF) by gentle agitation for 1 hr at 4 °C. Insoluble fractions were removed by centrifugation at 50,000 $g$ for 30 min and supernatant was incubated with 200 µl Neutravidin agarose slurry pre-equilibrated in wash buffer (25 mM HEPES, pH 7.4, 200 mM NaCl, 5% glycerol, 0.02% GDN) for 1 hr at 4 °C. The resin was washed three times with 200 µl of wash buffer and surface-biotinylated proteins were eluted by incubation with 200 µl wash buffer containing 50 mM fresh DTT for 1 hr at 4 °C with gentle mixing of the sample. Twenty µl of input, flow-through and surface-eluted samples were separated by SDS-PAGE. Samples were then transferred to a PVDF membrane and analyzed by Western blot using a mouse-anti-myc primary antibody (Sigma) and HRP-coupled goat-anti-mouse secondary antibody (Jackson ImmunoResearch). Antibodies were diluted 1:10,000 in TBS-T with 5% skimmed milk. Chemoluminescence signal was developed using ECL prime reagent (Cytiva) and imaged using a Fusion FX7 imaging system (Vilber).

## Liposome reconstitution

For liposome reconstitutions using the [Eu.L1$^+$] bicarbonate-selective probe, lipid preparation was performed on the same day as protein purification. POPC:Cholesterol lipids (7:3 w/w ratio, Avanti) were pooled in a round-bottom flask and initially dried under $N_2$ flow to form a lipid film. Lipids were further dried under vacuum for 1 hr prior to re-hydration with [Eu.L1$^+$] buffer (50 µM [Eu.L1$^+$], 200 mM NaCl, 5 mM HEPES, pH 7.4). The resulting suspension was gently sonicated and stirred for 2 hr to generate a homogeneous mixture. Ten freeze-thaw cycles were performed to generate multilamellar liposomes before extrusion 29 times through x2 polycarbonate 400 nm filters to give unilamellar liposomes.

For protein incorporation, 10 μl aliquots of 10% Triton X-100 were added in order to destabilize liposomes and permit protein incorporation. After reaching a plateau of the light scattering measured at 540 nm, 4 additional aliquots of Triton X-100 were added. The number of additions required for destabilization did not vary between reconstitutions. After the formation of destabilized liposomes, purified SLC26A6 was added at a lipid-to-protein ratio (LPR) of 50:1. To counteract [Eu.L1$^+$] probe leakage, an additional equimolar 50 μM [Eu.L1$^+$] was added. The subsequent mixture was gently rotated at RT for 20 min prior to the addition of 250 mg SM-2 Bio-Beads (Bio-Rad) per 5 ml of sample. The sample was then incubated for 30 min at RT prior to incubation at 4 °C. Bio-Bead additions were performed over a 3-day period every 24 hr. Biobeads were removed by gravity filtration and the sample was used immediately for assay measurements.

For liposome reconstitution for the Lucigenin and ACMA assays, a similar procedure was used with the following adjustments. POPE:POPG lipids (3:1 w/w) were first pooled in a round-bottom flask and washed with diethyl ether prior to drying under $N_2$ for 1.5 hr. Lipids were then resuspended to 20 mg/ml in 10 mM HEPES, 150 mM KCl prior to sonication to form a homogeneous mixture. After three freeze-thaw cycles, the sample was frozen in liquid $N_2$ and stored at –80 °C until further use. Purified SLC6A9$^T$ was incorporated utilizing the triton-destabilisation method previously detailed at a LPR of 80:1 (w/w). Both SLC26A6 wildtype and the R404V mutant were incorporated at a 50:1 (w/w) ratio. Successful incorporation was confirmed by SDS-PAGE gel quantification. For all reconstitution procedures, mock empty liposomes lacking any purified protein were generated using the same protocols.

## [Eu.L1$^+$] bicarbonate transport assay

For all measurements of $HCO_3^-$ transport, buffers were degassed to remove atmospheric $CO_2$. After Bio-Bead removal via gravity filtration, proteoliposomes were pelleted and the excess [Eu.L1$^+$] probe (*Martínez-Crespo et al., 2021*) was removed by two ultracentrifugation and resuspension steps in symmetric outside buffer (200 mM NaCl, 5 mM HEPES, pH 7.4). Where 50 mM external NaCl was required, liposomes were pelleted by ultracentrifugation and resuspended in symmetric internal buffer (50 μM [Eu.L1$^+$], 200 mM NaCl, 5 mM HEPES, pH 7.4). Liposomes were further subjected to five freeze-thaw cycles and extruded 29 times using 400 nm polycarbonate filters. Two ultracentrifugation and resuspension-wash steps in asymmetric external buffer (50 mM NaCl, 5 mM HEPES, pH 7.4) were performed before the final resuspension in asymmetric external buffer.

For measurements of electroneutral $Cl^-$/ $HCO_3^-$ exchange, 100 μl of SLC26A6 proteoliposomes were added to a 96-well flat bottom black plate (Greiner). Prior to measurement, liposome samples were gently bubbled under $N_2$ for 2 min to remove atmospheric $CO_2$ and residual $HCO_3^-$ from samples. Following the recording of a baseline emission for 20 fluorometric excitation/emission cycles (lasting 0.5 s each) using a Tecan Infinite M1000 Pro microplate reader, 10 mM $HCO_3^-$ was added to the liposomes. Normalized [Eu.L1+] emission (excitation = 332, emission = 617) was determined following the lysis of liposomes by addition of 0.4% Triton X-100 after cycle 300. Time gating of the recorded emission was performed using a 150 μs lag-time between excitation and signal integration with an overall 850 μs integration time. $HCO_3^-$ transport was measured in the presence of both, a four-fold NaCl gradient (200 mM inside/50 mM outside) and symmetrical conditions (200 mM inside/200 mM outside).

## ACMA assay

To compare the uncoupled chloride conduction properties of SLC26A6 and SLC26A9, proteoliposomes were prepared with an internal buffer concentration of 50 mM $Na_2SO_4$, 10 mM HEPES, pH 7.4. Proteoliposomes were sonicated in 20 μl aliquots to form unilamellar vesicles prior to dilution into 9-amino-6-chloro-2-methoxyacridine (ACMA) assay buffer (2 μM ACMA, 75 mM NaCl, 10 mM HEPES, pH 7.4). 100 μl aliquots were added to a 96-well plate and after a baseline measurement period for 13 excitation/emission cycles (0.5 s per cycle), the protonophore carbonyl cyanide 3-chlorophenylhydrazone (CCCP) was added to dissipate the membrane potential and permit transporter-mediated anion-movement. ACMA fluorescence (excitation = 412 nm and emission = 482 nm), as a determinant of $Cl^-$ transport mediated $H^+$ influx, was measured using the Tecan Infinite M1000 Pro Plate Reader over 495 s or using the Tecan Spark Plate reader over 1800 s. Data were normalized to the point of CCCP addition.

To measure electrogenic oxalate transport, the same protocol was applied with the following exception: SLC26A6 liposomes consisted of an internal concentration of 150 mM KCl, 10 mM HEPES, pH 7.4 and an external concentration of 9.4–150 mM oxalate, 10 mM HEPES, pH 7.4. A total of 40 µl aliquots were added to a to a black 384-well plate (Greiner) and ACMA fluorescence was measured on a TECAN Spark Plate Reader over 1200 s.

## Lucigenin assay

For measurement of $Cl^-$ uptake as consequence of $Cl^-/HCO_3^-$ exchange, the halide-sensitive lucigenin dye (400 µM) was incorporated into proteoliposomes containing either SLC26A6, the mutant SLC26A6 R404V or mock liposomes (all with a POPE:POPG composition of 3:1 w/w) in lucigenin assay buffer (200 mM $HCO_3^-$, 10 mM HEPES, pH 7.4). To permit fluorophore incorporation, liposomes were diluted in the above buffer and three freeze-thaw cycles were performed prior to extrusion (17 times) using 50 nm polycarbonate filters. Ultracentrifugation was performed twice and resuspension in lucigenin assay buffer was performed to remove excess exterior lucigenin. Forty µl aliquots were transferred to a black 384-well plate and after 20 cycles (0.2 s per cycle), 0–240 mM NaCl diluted in assay buffer was added. Quenching of lucigenin fluorescence (excitation = 430 nm, emission = 505 nm) was measured as a determinant of $Cl^-$ influx for 300 cycles prior to lysis by 0.4% Triton X-100. Data are normalized to the point of NaCl addition and corrected for signal observed under mock liposome conditions.

## Electrophysiology

For electrophysiology recordings, adherent HEK293T cells were maintained in DMEM (Gibco) media and incubated at 37 °C and 5% $CO_2$. Media was supplemented with 10% FBS, 1 mM L-glutamine, 4.5 g $l^{-1}$ Glucose, 1 mM sodium pyruvate and 100 U $ml^{-1}$ penicillin/streptomycin (Sigma). Cells were transfected at between 40% and 60% confluency and split into 60x15 mm dishes (Corning) the day prior to transfection. 5 µg plasmid DNA was added together with linear PEI (25 kDa) at a ratio of 1:2.5 (w/w) DNA to PEI for transfection. DNA and PEI were mixed in non-supplemented media, incubated at RT for 10 min and added dropwise to cells with 3 mM VPA. Venus-fused constructs of human SLC26A6 and mouse SLC26A9 were used for recordings to facilitate the detection of expression through fluorescence microscopy. Cells were recorded from between 16 and 30 hr post-transfection.

Patch pipettes were formed from pulled and polished borosilicate glass capillaries (OD = 1.5 mm, ID = 0.86 mm, Sutter). When backfilled with 150 mM CsCl, patch pipettes yielded a resistance of between 2–4 MΩ. For voltage-clamp experiments, currents were recorded with an Axopatch 200B amplifier and digitized using a Digidata 1440 A A/D converter. Signals were filtered at 5 kHz and sampled at 20 kHz prior to acquisition with Clampex 10.6 (Molecular Devices). A voltage step protocol using a holding potential of 0 mV for 0.2 s was performed prior to 20 mV incremental voltage steps. The voltage steps ranged from –100 mV to +100 mV before returning to 0 mV. Recorded cell capacitance upon break-in varied between 10–30 pF with series resistance <10 MΩ. Calculated liquid junction potentials (JPCalcW, Molecular Devices) never surpassed 5 mV.

For whole-cell patch-clamp experiments, the intracellular pipette solution consisted of 146 mM CsCl, 2 mM $MgCl_2$, 5 mM EGTA, 10 mM HEPES, pH 7.4. Extracellular bath solutions consisted of either 146 mM CsCl, 2 mM $MgCl_2$, 5 mM EGTA, 10 mM HEPES, pH 7.4 or 150 mM $CsHCO_3$/150 mM $Cs_2Oxalate$, 2 mM Mg(OAc)$_2$, 5 mM EGTA, 10 mM HEPES with pH adjusted using either CsOH or methanesulfonate. The pH of $HCO_3^-$ was monitored to ensure that the production of $CO_2$ and the consequent decline of $HCO_3^-$ never exceeded 20%. All acquired electrophysiology data were analyzed using Clampfit 11.0.3 (Molecular Devices), Excel (Microsoft), and GraphPad Prism 9 (GraphPad).

## Cryo-EM data acquisition

To proceed with structure determination by cryo-EM, 2.5 µl samples of GDN-purified SLC26A6 were applied to glow-discharged holey carbon grids (Quantifoil R1.2/1.3 Au 200 mesh) at a concentration of 2 mg $ml^{-1}$. Blotting of the sample (3–6 s) was performed at 4 °C at a relative humidity of 75% and grids were flash-frozen in liquid propane-ethane mix with a Vitrobot Mark IV (Thermo Fisher Scientific). A total of 11,962 images from two combined datasets were collected on a 300 kV Titan Krios G3i using a 100 µm objective aperture. Data were collected in super-resolution mode using a post-column quantum energy filter (Gatan) with a 20 eV slit and a K3 Gatan direct detector. Data were acquired using EPU 2.7 for dataset 1 and EPU 2.9 for dataset 2 with aberration-free image shift (AFIS)

and a defocus range of –0.8 to –2.4 μm. The dataset was recorded using a pixel size of 0.651 Å/pixel (0.3225 Å/pixel in super-resolution mode) with a total exposure time of 1 s (36 individual frames) and a dose of 1.696 e⁻/Å²/frame for dataset 1 and 1.85 e⁻/Å²/frame for dataset 2. The total electron dose for the specimen was 61 e⁻/Å² for dataset 1 and 67 e⁻/Å² for dataset 2.

### Cryo-EM data processing

Cryo-EM data were processed using CryoSPARC v3.2.0–4.0 (*Punjani et al., 2017*; *Figure 2—figure supplement 1*, *Table 1*). All movies were subjected to patch motion correction. Following patch CTF estimation, high quality micrographs were identified based on relative ice thickness, CTF resolution estimation and total full frame motion. 1,749,907 particles were picked following the generation of 2D templates for automated template picking. Subsequently, picked particles were binned 2.4 x using a box-size of 360 pixels for 2D classification (pixel size 1.56 Å/pixel). Following three rounds of 2D classification, classes were selected that displayed general structural features characteristic for the SLC26 family in various orientations. From this, 256,581 particles were used to generate a 'good' ab initio reconstruction and a portion of rejected particles (16,030 particles) were used to generate a 'junk' reconstruction. 967,713 selected particles from 2D classification were subjected to heterogeneous refinement using the selected good reconstruction as a 'template' and the junk reconstruction as a 'decoy'. After several rounds of heterogenous refinement, the selected 507,174 particles and associated heterogeneously refined map were subjected to multiple rounds of non-uniform refinement with imposed C2 symmetry. Following iterative rounds of non-uniform and local CTF refinement, a reconstruction at a nominal resolution of 3.55 Å was generated. These particles were then subjected to a four-class ab initio reconstruction for further sorting resulting in one good map composed of 173,834 particles. Particles were re-extracted with a bin-1 pixel size of 0.651 Å/pixel using a 432 pixel-sized box and subjected to non-uniform refinement prior to a final three-class ab initio reconstruction resulting in a single high-quality map comprised of 93,169 particles. Multiple rounds of non-uniform refinement and local CTF refinement were performed. Finally, the resultant reconstruction maps were sharpened with the DeepEMhancer tool using the HighRes deep-learning model (*Sanchez-Garcia et al., 2021*). The quality of the map was analyzed with 3DFSC (*Tan et al., 2017*) for FSC validation and local resolution estimation.

### Cryo-EM model building and refinement

Map interpretation was performed in Coot (*Emsley and Cowtan, 2004*; *Emsley et al., 2010*) using a model of SLC26A6 obtained from the AlphaFold Protein Structure Database (*Jumper et al., 2021*; *Varadi et al., 2022*) as template. The quality of the map allowed for the unambiguous assignment of residues 28–49, 62–594, and 655–747. The model was iteratively improved by real space refinement in PHENIX (*Afonine et al., 2018*; *Liebschner et al., 2019*) maintaining NCS and secondary structure constrains throughout. Figures were generated using ChimeraX (*Pettersen et al., 2004*; *Pettersen et al., 2021*) and Dino (http://www.dino3d.org). Surfaces were generated with MSMS (*Sanner et al., 1996*).

## Acknowledgements

We thank the Center for Microscopy and Image Analysis (ZMB) of the University of Zurich for the support and access to the electron microscopes and Anastasiia Sukalskaia, Elena Lehmann, Katarzyna Drozdzyk and Melanie Arndt for their help in cryo-EM data acquisition. All members of the Dutzler lab are acknowledged for their help at various stages of the project. Justin Walter is acknowledged for the cloning of initial constructs and advice in the SLC26 project. This research was supported by a grant from the Swiss National Science Foundation (No. 31003A_163421) to RD. SJB acknowledges the Engineering and Physical Sciences Research Council (EPSRC) for funding [EP/S032339/1].

# Additional information

## Funding

| Funder | Grant reference number | Author |
|--------|------------------------|--------|
| Schweizerischer Nationalfonds zur Förderung der Wissenschaftlichen Forschung | 31003A_163421 | Raimund Dutzler |
| Engineering and Physical Sciences Research Council | EP/S032339/1 | Stephen J Butler |

The funders had no role in study design, data collection and interpretation, or the decision to submit the work for publication.

## Author contributions

David N Tippett, Conceptualization, Data curation, Formal analysis, Validation, Investigation, Visualization, Methodology, Writing – original draft, Writing – review and editing, Cloned, expressed and purified proteins, carried out electrophysiology and transport experiments, prepared samples for cryo-EM, processed cryo-EM data and built models; Colum Breen, Resources, Writing – review and editing, Synthesized probe used for HCO3- transport experiments; Stephen J Butler, Resources, Writing – review and editing; Marta Sawicka, Data curation, Formal analysis, Supervision, Writing – review and editing, Assisted in cryo-EM data collection and processing; Raimund Dutzler, Conceptualization, Data curation, Formal analysis, Supervision, Funding acquisition, Validation, Visualization, Writing – original draft, Project administration, Writing – review and editing

## Author ORCIDs

Marta Sawicka ⬥ http://orcid.org/0000-0003-4589-4290
Raimund Dutzler ⬥ http://orcid.org/0000-0002-2193-6129

Reviewer #1 (Public Review): https://doi.org/10.7554/eLife.87178.3.sa1
Reviewer #2 (Public Review): https://doi.org/10.7554/eLife.87178.3.sa2
Reviewer #3 (Public Review): https://doi.org/10.7554/eLife.87178.3.sa3
Author Response: https://doi.org/10.7554/eLife.87178.3.sa4

# Additional files

## Supplementary files

• Transparent reporting form

## Data availability

The cryo-EM density map of hSLC26A6 has been deposited in the Electron Microscopy Data Bank under ID code EMD-17085. The coordinates for the atomic model of hSLC26A6 have been deposited in the Protein Data Bank under ID code 8OPQ. Source data files have been provided for Figure 1, Figure 1-figure supplement 1, Figure 2-figure supplement 1 and Figure 5.

The following datasets were generated:

| Author(s) | Year | Dataset title | Dataset URL | Database and Identifier |
|-----------|------|---------------|-------------|-------------------------|
| Tippett DN, Breen C, Butler SJ, Sawicka M, Dutzler R | 2023 | Structure of Human Solute Carrier 26 family member A6 (SLC26A6) anion transporter in an inward-facing state | https://www.rcsb.org/structure/8OPQ | RCSB Protein Data Bank, 8OPQ |

*Continued on next page*

*Continued*

| Author(s) | Year | Dataset title | Dataset URL | Database and Identifier |
|---|---|---|---|---|
| Tippett DN, Breen C, Butler SJ, Sawicka M, Dutzler R | 2023 | Structure of Human Solute Carrier 26 family member A6 (SLC26A6) anion transporter in an inward-facing state | https://www.ebi.ac.uk/emdb/EMD-17085 | Electron Microscopy Data Bank, EMD-17085 |

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
