## [Editor Report · eLife assessment]

This **important** manuscript combines cryo-EM and a suite of **compelling** whole cell and proteoliposome transport assays to establish the mechanism and structure of the full-length human SLC26A6 chloride/bicarbonate exchangers, including the first partial view of the previously unresolved IVS region of an SLC26 STAS domain. In combination with prior studies on additional SLC26 paralogs, including the SLC26A9 paralog initially reported by the same group, the study provides broadly relevant insights into the mechanistic diversity of the SLC26 transporters. This study is of interest to the biophysics community and the field of membrane transport.

---

## [Referee Report · Reviewer #1 (Public Review)]

Tippett et al present whole cell and proteoliposome transport data showing unequivocally that purified recombinant SLC26A6 reconstituted in proteoliposomes mediates electroneutral chloride/bicarbonate exchange, as well as coupled chloride/oxalate exchange unassociated with detectable current. Both functions contrast with the uncoupled chloride conductance mediated by SLC26A9. The authors also present a novel cryo-EM structure of full-length human SLC26A6 chloride/anion exchanger. As part of the structure, they offer the first partial view of the STAS domain previously predicted to be unstructured. They further define a single Arg residue of the SLC26A6 transmembrane domain required for coupled exchange, mutation of which yields apparently uncoupled electrogenic chloride transport mechanistically resembling that of SLC26A9, although of lower magnitude. The authors further apply to proteoliposomes for the first time a still novel approach to the measurement of bicarbonate transport using a bicarbonate-selective Europium fluorophor. The evidence strongly supports the authors' claims and conclusions, with one exception.

The manuscript has numerous strengths:

As a structural biology contribution, the authors extend the range of SLC26 structures to SLC26A6, comparing it in considerable detail to the published SLC26A9 structure, and presenting for the first time the structure of a portion of the STAS IVS domain of SLC26A6 long considered unstructured.

The authors also apply a remarkably extensive range of creative technical approaches to assess the functional mechanisms of anion transport by SLC26A6, among them the first application of the novel, specific bicarbonate sensor Eu-L1+ to directly assess bicarbonate transport in reconstituted proteoliposomes. The authors also present the first (to this reviewer's knowledge) functional proteoliposome reconstitution of chloride-bicarbonate exchange mediated by an SLC26 protein. They define a residue in surrounding the anion binding pocket which explains part of the difference in anion exchange coupling between SLC26A6 and SLC26A9. In the setting of past conflicting results, the current work also contributes to the weight of previous evidence demonstrating that SLC26A6 mediates electroneutral rather than electrogenic Cl-/HCO3- exchange.

Each of these achievements constitutes a significant advance in our understanding.

The paper has only a few weaknesses:

One is an incomplete explanation of the mechanistic determinants of anion exchange coupling in SLC26A6 vs. uncoupled anion transport by SLC26A9.

A second weakness is the inconsistent, technique-dependent detection of SLC26A6- mediated electrogenic chloride/oxalate exchange. In particular whole cell currents attributable to SLC26A6 in SLC26A6-expressing HEK293 cells in an oxalate bath could not be detected, whereas robust, saturable Cl- efflux into oxalate solution from proteoliposomes reconstituted with recombinant SLC626A6 was detectable by AMCA fluorescence decay. This discrepancy was attributed to the relative sensitivities and/or signal-to-noise ratios of the assays.

Overall, the manuscript represents an important advance in our understanding of the SLC26 protein family and of coupled vs uncoupled carrier-mediated anion transport.

---

## [Referee Report · Reviewer #2 (Public Review)]

The eleven paralogs of SLC26 proteins in humans exhibit a remarkable range of functional diversity, spanning from slow anion exchangers and fast anion transporters with channel-like properties, to motor proteins found in the cochlear outer hair cells. In this study, the authors investigate human SLC26A6, which functions as a bicarbonate (HCO3-)/chloride (Cl-) and oxalate (C2O42-)/Cl- exchanger, combining cryo-electron microscopy, electrophysiology, and in vitro transport assays. The authors provide compelling evidence to support the idea that SLC26A6's exchange anions at equimolar stoichiometry, leading to the electroneutral and electrogenic transport of HCO3-/ Cl- and C2O42-/Cl-, respectively. Furthermore, the structure of SLC26A6 reveals a close resemblance to the fast, uncoupled Cl- transporter SLC26A9, with the major structural differences observed within the anion binding site. By characterizing an amino acid substitution within the SLC26A6 anion binding site (R404V), the authors also show that the size and charge variance of the binding pocket between the two paralogs could, in part, contribute to the differences in their transport mechanisms.

This is a well-executed study, and the strength of this work lies in the reductionist, in vitro approach that the authors took to characterize the transport process of SLC26A6. The authors used and developed an array of functional experiments, including two electrogenic transport assays - a fast kinetic (electrophysiology) and a slow-kinetic (fluorescent-based ACMA) - and two electroneutral transport assays, probing for Cl- (lucigenin) and HCO3- (europium), which are well executed and characterized. The structural data is also of high quality and is the first structure of an SLC26 coupled anion exchanger, providing essential information for clarifying our understanding of the functional diversity between the SLC26 family of proteins.

---

## [Referee Report · Reviewer #3 (Public Review)]

The mechanistically diverse SLC26 transporters play a variety of physiological roles. The current manuscript establishes the SLC26A6 subtype as electroneutral chloride/bicarbonate exchanges and reports its high-resolution structure with chloride bound.

The claims in this manuscript are all well-supported by the data. Strengths include the comprehensive functional analysis of SLC26A6 in reconstituted liposome vesicles. The authors employ an array of assays, including chloride sensors, a newly developed fluorescent probe for bicarbonate, and assays to detect the electrogenicity of anion exchange. With this assortment of assays, the authors are able to establish the anion selectivity and stoichiometry of SLC26A6. Another strength of the manuscript is the functional comparison with SLC26A9, which permits fast, passive chloride transport, in order to benchmark the SLC26A6 activity. The structural analysis, including the assignment of the chloride binding site, is also convincing. The structural details and the chloride binding site are well-conserved among SLC26s. Finally, the authors present an interesting discussion comparing the structures of SLC26A5, SLC26A6, and SLC26A9, and how the details of the chloride binding site might influence the mechanistic distinctions between these similar transporters.

---

## [Author Response]

The following is the authors' response to the original reviews.

We thank the reviewers for their positive remarks, which we have addressed in detail below and which we have considered in our revised manuscript.

**Reviewer #1 (Recommendations For The Authors):**
The authors claim several times to have documented electrogenic chloride/oxalate exchange mediated by human SLC26A6. However, they fail to detect whole cell currents in SLC26A6-expressing HEK293 cells in oxalate bath, despite robust, saturable Cl- efflux from proteoliposomes into extracellular oxalate solution, as detected by AMCA fluorescence decay.

We interpret the low, and essentially non-detectable currents for Cl-/oxalate exchange as a consequence of the slow kinetics of transport. This lack of sensitivity is not unusual for electrogenic secondary-active transport processes recorded by patch-clamp electrophysiology in mammalian cells, which renders the recording in large *X. laevis* oocytes by two-electrode voltage clamp the preferred method for such investigations. In contrast to the non-detectable activity in electrophysiology, the pronounced signal in the ACMA assay reflects the influx of H+ as a consequence of the negative membrane potential established by the influx of the divalent anion oxalate, which we assume to occur in exchange with the monovalent Cl-.

Instances in the manuscript include:Abstract Line 17 overstates the paper's findings as "we have characterized SLC26A6 as a strictly coupled exchanger of chloride with either bicarbonate or oxalate". To the extent that "strictly coupled" implies 1:1 stoichiometry, the authors conclude Cl-/bicarbonate exchange is electroneutral based on its lack of exchange current. In contrast, the lack of Cl/oxalate exchange current does not lead the authors to the same conclusion of electroneutrality for Cl-/oxalate exchange. The data presented do not measure the stoichiometry of Cl-/oxalate exchange.

We agree that our ACMA experiments do not strictly discriminate between coupled and uncoupled oxalate transport. However, it should be emphasized that, assuming that transport proceeds by an alternate access mechanism, uncoupled oxalate transport would require the change of the unloaded transporter between inward- and outward-facing conformations, which was shown to be unfavorable in Figure 1D.

We have reworded the sentence in the abstract to:

“Here we have determined the structure of the closely related human transporter SLC26A6 and characterized it as a coupled exchanger of chloride with bicarbonate and presumably also oxalate.”

Line 264 claims that "the paper's functional data has defined SLC26A6 as a coupled transporter that exchanges Cl- with either HCO3- or oxalate at equimolar stoichiometry."

We have changed the sentence to:

“Whereas our functional data has defined SLC26A6 as a coupled antiporter that exchanges Cl- with HCO3- and presumably also oxalate with equimolar stoichiometry…”

In lines 299-302, the authors claim to have "detected strict equimolar exchange of anions"...leading to the reasonable conclusion of electroneutral Cl-/HCO3- exchange and the reasonable but unsupported conclusion of coupled Cl/oxalate exchange.

We have reworded the sentence to:

“In the case of Cl-/HCO3- transport, we detect a strict equimolar exchange of anions binding to a conserved site in the mobile core domain of the transmembrane transport unit (Figure 4B, H). Although not shown unambiguously, we assume an analogous mechanism also for Cl-/oxalate exchange.”

Lines 505-508 in Methods claim that the AMCA proteoliposome assay "measured electrogenic oxalate transport." However, the assay documented extracellular oxalate- dependent anion transport that was most simply interpreted as coupled exchange.

The assay has detected H+ uptake into proteoliposomes as a consequence of electrogenic anion influx. In these experiments, oxalate is the only anion on the outside of vesicles and it requires to be transported to be able to observe any fluorescent change. The claim of electrogenic oxalate transport is thus justified. As described above, the assay does under the applied conditions not discriminate between uncoupled and coupled oxalate transport, however uncoupled oxalate transport would require the conformational change of an unloaded transporter, which was shown to be kinetically disfavored.

In contrast, other parts of the manuscript acknowledge that the evidence presented falls short of documenting stoichiometric chloride/oxalate exchange.Results Line 151 sets out to "investigate a potentially electrogenic Cl-/oxalate exchanger. Similarly, results line 160 conservatively claims that Cl-/oxalate exchange occurs "presumably" with a 1:1 stoichiometry. This more accurate language needs to be used throughout the paper, replacing the more absolute but unjustified descriptions summarized earlier above.

We have now introduced the requested clarifications throughout.

I have otherwise only Minor points to suggest. Abstract:"Among the eleven paralogs in humans.... ". This should be "at least 10," as the originalstatus of human SLC26A10 as a transcribed pseudogene vs. a truncated protein-expressing gene remains unresolved. The authors recognize this in the introduction, where on p. 3 they acknowledge "ten functional SLC26 paralogs in humans."

We have changed to ‘ten functional paralogs’

Introduction:p. 4 line 45: membrane-inserted

We have introduced the correction.

Methods:Construct Generation:p. 25 lines 380-2: Add a sentence describing any C-terminal sequence extension added after C3 cleavage product, and whether/how it modified the PDZ-binding domain sequence. Has the modification been tested for PDZ-binding activity?

We have introduced the following sentence:

“As a consequence of FX cloning, expressed constructs include an additional serine at the N- terminus and an alanine at the C-terminus. Following C3-cleavage, SLC26A6 carries a seven residues long C-terminal extension (of sequence ALEVLFQ).”

We have not tested PDZ-domain binding but expect that the added residues interfere with interaction with the C-terminal binding motif.

Liposome Reconstitution:p.28: lines 453-4: Please clarify the meaning of: "absorbance at 540 nm was used to detect liposome destabilization," followed immediately by "After the formation of stabilized liposomes".... Does destabilization mean liposomal leak of Eu.L1+ chromophore, with decline of absorbance? What is practically meant in terms of the number of 10 mL additions of 10% TTX-100 routinely added to generate stabilized liposomes without generating destabilized liposomes? Did this number vary from trial to trial? How did you know when to stop adding aliquots of TTX-100?

We have added the following sentence:

“For protein incorporation, 10 µl aliquots of 10% Triton X-100 were added in order to destabilize liposomes and permit protein incorporation. After reaching a plateau of the light scattering measured at 540 nm, 4 additional aliquots of Triton X-100 were added. The number of additions required for destabilization did not vary between reconstitutions.”

p.28 line 463: "dissolved" should be "suspended."

We have introduced the correction.

Bicarbonate Transport Assayp. 29 line 480-1. How many repetitions represented by the phrase "sequential ultracentrifugation steps"- please provide a number or a range, as applicable.

We have defined the number of ultracentrifugation steps (two).

Pp 29-30, lines 485-7: define "cycles" - are these fluorometric excitation-emission cycles?

We have defined cycles as fluorometric excitation/emission cycles.

p. 30 line 489: delete "by"

We have deleted ‘by’

Name the fluorimeter used.

We have named the fluorimeter used as Tecan Infinite M1000 Pro microplate reader.

AMCA assayPp 30-31, lines 505-8: Add composition of extraliposomal oxalate-containing buffer. In Fig 1 Suppl Fig. 1 panels H and I, and Methods lines 505-508, with 150 mM oxalate substituting for 150 mM Cl- how was osmotic balance maintained in the external chloride solution?

We have added the composition of the oxalate-containing buffer. The osmolarity of the extracellular solution was not balanced.

Electrophysiologyp.32 line 532: What fold-increase of SLC26 protein levels was produced by inclusion of 3 mM valproic acid?

We consistently see an increase of expression upon addition of valproic for different membrane proteins acid but did not quantify it in this case.

Results:Functional characterization of SLC26A6Line 91: "comparably" to what? Otherwise, perhaps, "comparatively" was intended here?

We have changed to ‘comparatively’

Fig 1E legend: line 763 "time- and concentration-dependent". Same for line 791, line 799

We have introduced the correction.

Fig. 1G: Change Y axis legend to "Normalized [Eu.L1+] emission." Add bath ion composition for "neg" condition (black trace).

We have corrected the label on the Y-axis and added ion composition for neg.

Fig. 1H legend sentence 2 "in a concentration-dependent manner for liposomes (Mock) in75 mM oxalate (n=5) and for SLC26A6 proteoliposomes in extracellular oxalate concentrations of 9.4 mM (n=3) etc

We have reworded the sentence:

“Traces show mean quenching of ACMA fluorescence in a time- and concentration-dependent manner for SLC26A6 proteoliposomes with outside oxalate concentrations of 9.4 mM (n = 3), 37.5 mM (n = 5), 75 mM (n = 6), 150 mM (n = 8, all from two independent reconstitutions). Neg. refers to liposomes not containing SLC26A6 assayed upon addition of 75 mM oxalate as defined in Figure1-figure supplement 1G.”

Fig. 1 Fig Suppl. 1. p.45 line 790: change "chemical formulas" to "2-D chemical structures"

We have introduced the change.

lines 799: Time-

We have introduced the change.

Fig 1 Fig Suppl. 1. p 46 lines 809-810: dashed lines indicating 0 pA are indeed red in panels A and B, but black in panels H and I.

We explicitly refer to recordings, where dashed lines at 0 pA are consistently in red.

Fig 2 Fig Suppl.1 p. 50, line 832: Two additional multi-class.

We have introduced the change.

Fig 2 Suppl Fig 2B, p. 51: Please label the residue numbers of the side chains coordinating the chloride binding site. Can those residues be indicated in Fig 2 Suppl Fig 2A in the appropriate helices? These residues might also be asterisked in the primary sequence alignment of Fig 2 Suppl Fig 3A.

We have labeled the residues in Fig. 2-figure supplement 2B but not 2A where the focus is on the general quality of the density in different parts of the protein. We have also labeled the same residues in Fig. 2-figure supplement 3A.

Fig.4 legend p. 59 line 882 – Deviating residues in SLC26A9 (typo A6) are highlighted in violet.

We have introduced the correction.

p. 60 lines 888-9: Please clarify the individual meaning of green and purple asterisks on defining the substrate cavity diameter; How do purple and green asterisks relate to the yellow and green lines in the graph? Should the asterisks be two green and two yellow asterisks, or should they be black? What is the meaning of the purple and green asterisks at the two upper corners of panel G with respect to the substrate cavity radii?Please specify if y axis label "radius" refers to substrate cavity radius, and whether X axis label "distance" refers to axial distance along helix alpha10, alpha1, or of the helical pair. Is value "0 A" on the X axis anchored at the top of the helices as depicted in panels D-F? Is X-axis value 10.5A sited at the bottom of the helices? Please indicate on the panel G curves the x-y value range depicted in the inset images- or clarify that the inset images present the entire curves of panel G.

We have clarified these remarks in a revised legend:

“The radius of the substrate cavity of either protein is mapped along a trajectory connecting a start position at the entrance of each cavity (distance 0 Å) and an end position located outside of the cavity in the protein region (distance 10 Å). Both points are defined by asterisks in insets showing the substrate cavities for either transporter and they are indicated in the graph (green, cavity entrance towards the aqueous vestibule; violet, protein region).”

Fig. 4 p . 61-2 panel H and lines 907-8: Addition to the panel of the A5 "buried Cl- binding site" would be helpful, if possible to do without obscuring the A6 and A9 Cl-s.

Panels H and I show the cavity harboring the ion binding site in two orientations, including the surrounding residues. We prefer to show all surrounding residues for both orientations, even if this somewhat obscures the view on the ion in the left panels. An unobscured view of the ion in its cavity is provided in panels D and E.

p. 12. Results line 234: Please specify that "both proteins" here refers to A6 and A5 vs A6 and A9

We have specified this.

p. 13 Results lines 268-72: R404 is "ubiquitous in other mammalian paralogs..." should be changed to "shared by most but not all mammalian paralogs".

We have changed the text accordingly.

Fig 4C should have a red or purple asterisk placed under the yellow column corresponding to R404 of SLC26A6, so that the discussion can refer to it. It would also be helpful to remind the reader that R404 corresponds to conserved position 6 in Fig 4 Fig Suppl. 1 panels D-G.Here the authors might note that sulfate -transporting SLC26A1 and -A2 have the shorter side chain K residue.

We have marked the position in Figure 4C with an asterisk and added the following sentence to its legend:

Asterisk marks position that harbors a basic residue in all family members except for SLC26A9 where the residue is replaced by a valine. Whereas most paralogs, including the ones operating as bicarbonate exchangers, have an arginine at this site, the sulfate transporters SLC26A1 and 2 contain a smaller lysine.

We have added the following statement to the legend of Figure4-figure supplement 1:

“‘6’ indicates the position which contains a basic residue in all family members except for SLC26A9.”

Fig 5B legend p. 63 line 916. Please specify if the 14 independent experiments include both the symmetric Cl- conditions and the asymmetric Cl-/HCO3- conditions or only one condition.

The 14 independent experiments were only recorded in symmetric chloride conditions. We have changed the legend accordingly.

Fig 5C. It would be useful for readers to add the I-V trace of WT SLC26A6 taken from Fig 1 Suppl 1B (perhaps in gray), to document the specificity of the very low magnitude R404V whole cell current. Alternatively, please note (if the case) that WT SLC26A6 currents (Fig 1 Supple 1B) are indistinguishable from the blacked dashed zero current density line.

We have now displayed the I-V trace of WT SLC26A6 as grey dashed line for comparison and added a new panel that show the differences between the currents of R404V and WT recorded at 100 mV (Fig. 5D). Although the currents for R404V were consistently lager than for WT, the difference is not statistically significant. We have explicitly mentioned this in the text and the figure legend.

Fig 5E depiction of decline in ACMA fluorescence is missing from the legend. Legend references to panels E and F seem to correspond to Fig 5 panels F and G (lucigenin fluorescence), as noted in Results p 14 lines 280-3.

We have added the legend.

Chernova et al (2005) reported electroneutral human and mouse A6-mediated Cl/HCO3- exchange in *Xenopus* oocytes. They also observed electrogenic Cl-/oxalate exchange by mouse SLC26A6, but detected no current generated by human SLC26A6-mediated Cl-/Oxalate exchange. That paper (already cited) might be referred to more explicitly in connection to the authors' current findings of electroneutral Cl-/HCO3- exchange by human SLC26A6 as well as their inability to detect human SLC26A6-mediated Cl-/oxalate exchange current in HEK-293 whole cell recordings.

We now have included the reference in the discussion:

“Consequently, transport would be electroneutral in case of the monovalent HCO - and electrogenic in case of the divalent oxalate (Figure 1E-H), which was already proposed in a previous study (Chernova et al., 2005). We also want to re-emphasize that the inability to measure discernable currents does not necessarily imply that the transport might not be electrogenic as, due to their slow kinetics, transport-mediated currents might be below the detection limit of patch-clamp electrophysiology.”

**Reviewer #2 (Recommendations For The Authors):**
- It would be helpful if the authors briefly clarify the depiction/scheme of the hypothetical SLC26A6 outward-facing conformation. Is this gleaned from a prior structure of a related SLC family or distant homolog? Functional data? Biochemical/biophysical data? As well, I would also recommend labeling this within the figure (Figure 3-figure supplement 1D labeling, for instance - inward, hypothetical outward).

As mentioned in the legend of Figure 3-figure supplement 1D, the outward conformation is hypothetical. We have also now mentioned this in the title. The displayed outward- conformation was constructed by manually moving the area depicting the core domain relative to the fixed gate domain.

- Have the authors attempted to block SLC26A6-mediated transport with the addition of a known inhibitor, such as niflumic acid? I understand that this may be technically challenging, but it would strengthen the transport assay data, especially in Figure 5D with the ACMA assay testing the SLC26A6 R404V mutant.

We have not attempted to block the currents by addition of niflumic acid.

- It could be helpful to the reader to move the schematics in Figure 1-figure supplement 1C into Figure 1.

We have now displayed the schematics illustrating the principle of the respective transport assays next to the data in Figure 1, but kept Figure 1-figure supplement 1C for a more detailed description of the assays.

- Figure 3-figure supplement 1D legend, should be "hypothetical" instead of "hypothetic."

We have introduced the correction.

- I might consider coloring the Cl- ion something that is distinct from the model colors that are used in the figures (see Figure 4 and Figure 4-figure supplement 1). This would help to clarify Figure4-figure supplement 1H, where I believed that the Cl- ions at first were from the SLC26A6 model at first glance.

We have used the green color for chloride throughout the manuscript and would prefer to keep it that way for consistency.

- Labeling in Figure 5 legend (E, F) do not match the Figure (F,G). The description of the ACMA assay is absent from the figure legend (the real Figure 5E).

This has been corrected.

**Reviewer #3 (Recommendations For The Authors):**
None. This is a well-done manuscript and I have no further suggestions.